# BAMDP Shaping: a Unified Framework for Intrinsic Motivation and Reward Shaping

**Aly Lidayan, Michael Dennis, Stuart Russell**
University of California, Berkeley
{dayan,michael_dennis,russell}@cs.berkeley.edu

## Abstract

Intrinsic motivation and reward shaping guide reinforcement learning (RL) agents by adding pseudo-rewards, which can lead to useful emergent behaviors. However, they can also encourage counterproductive exploits, e.g., fixation with noisy TV screens. Here we provide a theoretical model which anticipates these behaviors, and provides broad criteria under which adverse effects can be bounded. We characterize all pseudo-rewards as reward shaping in Bayes-Adaptive Markov Decision Processes (BAMDPs), which formulates the problem of learning in MDPs as an MDP over the agent's knowledge. Optimal exploration maximizes BAMDP state value, which we decompose into the value of the information gathered and the prior value of the physical state. Psuedo-rewards guide RL agents by rewarding behavior that increases these value components, while they hinder exploration when they align poorly with the actual value. We extend potential-based shaping theory (Ng et al., 1999) to prove BAMDP Potential-based shaping Functions (BAMPFs) are immune to reward-hacking (convergence to behaviors maximizing composite rewards to the detriment of real rewards) in meta-RL, and show empirically how a BAMPF helps a meta-RL agent learn optimal RL algorithms for a Bernoulli Bandit domain. We finally prove that BAMPFs with bounded monotone increasing potentials also resist reward-hacking in the regular RL setting. We show that it is straightforward to retrofit or design new pseudo-reward terms in this form, and provide an empirical demonstration in the Mountain Car environment.

## 1 Introduction

A common approach to guiding RL agents in sparse reward settings is to add pseudo-rewards to the true rewards. A wide range of pseudo-rewards have been used successfully in complex environments (Pathak et al., 2017; Berseth et al., 2019; Hafner et al., 2023); more domain-agnostic terms are typically known as intrinsic motivation (IM) while reward shaping is often more task-specific (Schmidhuber, 1991; Dorigo & Colombetti, 1994; Barto et al., 2004). However, designing them is challenging, and they can affect performance in counter-intuitive ways leading to degenerate behaviors (Taiga et al., 2021; Clark & Amodei, 2016). For instance, rewarding accurate prediction of the next percept may cause an agent to sit forever in front of a blank wall (Rhinehart et al., 2021), while favoring states where the next percept is hard to predict causes an agent to get stuck watching a "noisy TV" that randomly flips channels, rather than encouraging exploration as one might expect (Burda et al., 2018).

Avoiding these behaviors requires understanding their root causes. We approach this by analyzing pseudo-rewards within a theoretical framework for how a rational RL agent *should* behave—the Bayes-Adaptive MDP (BAMDP) (Bellman & Kalaba, 1959; Martin, 1967), a generalization of Bayesian bandits (Gittins, 1979). In a BAMDP, the agent starts out not knowing what MDP it is operating in and learns more through experience. BAMDP states consist of the cumulative information gathered, i.e., the entire history $h_t$ of MDP states, actions, and rewards observed up to and including the current state of the MDP. An RL algorithm can be viewed as a *policy* mapping the BAMDP state to an action that updates it (Duff, 2002), and optimal RL algorithms maximize the expected BAMDP return, i.e., the expected return while exploring and learning.

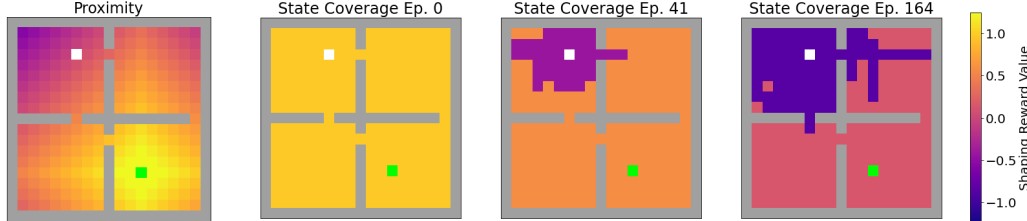

(a) $\phi(s) = -\|\mathbf{s} - \mathbf{g}\|_1$, i.e., the goal proximity.

(b) $\phi(h) = |\{s \in h\}|$, i.e., the count of all unique states visited so far. Over training, the agent must get further from $s_0$ to reach new states and increase $\phi(h)$.

Figure 1: Example potential-based reward shaping functions $\gamma\phi' - \phi$ for potentials over the MDP state (PBSF, 1a), and BAMDP state (BAMPF, 1b). Start and goal MDP states $s_0, g$ are marked white and green; other states are colored by the pseudo-reward for transitioning there directly from $s_0$ at the start of the episode. Each episode ends and $s$ is reset to $s_0$ every 50 steps, $\gamma = 0.99$.

Within this framework, we can understand both reward shaping and IM as *BAMDP reward shaping*, i.e., functions over BAMDP states, that guide exploration by incentivizing behavior that leads to more valuable BAMDP states. We decompose BAMDP state value into the value of the total information collected (VOI) and the value of the current MDP state under the prior, which we call the value of opportunity (VOO). This yields a new, principled typology of pseudo-reward terms based on the different types of BAMDP value that they signal. Many IM terms, e.g., prediction error (Pathak et al., 2017) or information gain (Houthooft et al., 2016), can be understood as incentivizing exploration by adding rewards to behavior that increases VOI. Other pseudo-rewards attempt to steer exploration by compensating for incorrect implicit VOO priors resulting from the initialization of RL algorithms. For example, goal proximity reward shaping (Colombetti et al., 1996) compensates for overly pessimistic VOO priors for states near the goal, while surprise minimization IM (Berseth et al., 2019) and joint angle violation penalties (Ji et al., 2023) compensate for overoptimistic VOO estimates for risky states. We can furthermore understand how pseudo-rewards mislead exploration as resulting from aligning poorly with actual value—e.g., watching a noisy TV yields high prediction error but no valuable information.

Next, we prove conditions under which pseudo-rewards cannot be "hacked" (where RL agents converge to final behaviors that maximize composite rewards to the detriment of the real rewards) in both RL and meta-RL settings. Potential-Based reward Shaping Functions (PBSFs) take the form $\gamma\phi(s') - \phi(s)$, where potential $\phi$ encourages going to higher value states by encoding their desirability, e.g., Fig. 1a. PBSFs on the BAMDP state (BAMPFs), e.g., Fig. 1b, may encode desirability of both the physical state *and* the total information gathered. Since they may depend on the entire training history, it is easy to convert most pseudo-rewards to BAMPFs. We extend results from Ng et al. (1999) to prove that BAMPFs always preserve the optimality of RL algorithms. Thus, we can use BAMPFs to guide meta-RL without it converging on RL agents that are optimal for the modified rewards but not the underlying domain. We empirically validate this by using a BAMPF to guide a meta-RL agent to learn an optimal RL strategy for a Bernoulli bandits domain. Next, we return to the regular RL setting and prove that BAMPFs based on potential functions that are bounded and monotone increasing will also preserve the optimal MDP policy at convergence, i.e., no RL algorithm can find a reward-hacking MDP policy maximizing shaped return but not real return. It is still straightforward to convert many pseudo-rewards into this form, which we illustrate with a case-study converting Curiosity (Pathak et al., 2017) to combat the Noisy TV problem. We finally demonstrate the practical utility of our framework in a larger-scale RL experiment. We use the BAMDP value decomposition to inform the design of an effective potential function for the continuous Mountain Car environment (Brockman et al., 2016), and show that the resulting BAMPF improves exploration while avoiding reward-hacking.

## 2 BACKGROUND

### 2.1 MARKOV DECISION PROCESSES

*Markov decision processes* (MDPs) are defined by tuple $M = (\mathcal{S}, \mathcal{A}, R, T, T_0, \gamma)$ with $\mathcal{S}$ a set of states, $\mathcal{A}$ a set of actions, $T(s'|s, a)$ and $R(r|s, a, s')$ transition and reward probability functions with expected reward $R(s, a, s')$, $T_0(s_0)$ an initial state distribution, and $\gamma$ a discount factor (when it is

not critical, we write $R$ without $s'$ for brevity). MDP policies map from current states to distributions over next actions: $\pi(a|s)$. The return is defined as the discounted sum of rewards $G = \sum_{t=0}^{\infty} \gamma^t r_{t+1}$, and an optimal policy $\pi^*$ maximizes the expected return.

## 2.2 INTRINSIC MOTIVATION AND REWARD SHAPING

*Intrinsic motivation* (IM) and *Reward Shaping* are methods for guiding RL algorithms by adding pseudo-rewards to the original reward at each step. IM functions $F(h_t)$ can depend on the entire history $h_t = s_0 a_0 r_1 s_1 ... a_{t-1} r_t s_t$ (the sequence of all transitions observed so far), thus IM can depend on RL algorithms' internal states, and in general the composite reward function $R(s_t, a_t, s_{t+1}) + F(h_{t+1})$ is not a valid MDP reward function. Reward shaping functions are restricted to the form $F(s, a, s')$, resulting in *shaped* MDPs with reward function $R'(s, a, s') = R(s, a, s') + F(s, a, s')$. Optimal policies for a shaped MDP, i.e., maximizing composite return, are generally not optimal for the original MDP. *Potential-based shaping functions* (PBSFs) take the form $F(s, a, s') = \gamma \phi(s') - \phi(s)$ and preserve optimal policies in all MDPs (Ng et al., 1999).

## 2.3 FORMULATION OF RL PROBLEMS AS BAMDPS

We formulate RL problems as BAMDPs, for which RL algorithms are policies. We use overlines, e.g., $\bar{M}$, to denote the BAMDP version of any object. Our conventions are inspired by Zintgraf et al. (2019) and Guez et al. (2012).

RL algorithms learn to maximize return in an MDP by repeatedly updating an internal state (e.g., a Q-function estimate) while selecting actions and receiving observations. We denote them by $\bar{\pi} : \mathcal{S} \times \mathcal{H} \rightarrow \mathcal{A}$, where $\mathcal{H}$ is the set of histories $h_t$ that are realizable, i.e., occur with non-zero probability for some action sequence (Abel et al., 2024). RL algorithms typically maintain a lossy memory of $h_t$, but it is a sufficient statistic for internal state, encapsulating all information $\bar{\pi}$ might use to choose actions. We measure the performance of $\bar{\pi}$ in MDP $M$ by its expected return *while learning*: $\mathcal{J}_M(\bar{\pi}) = \mathbb{E}_{M,\bar{\pi}}[\sum_{t=0}^{\infty} \gamma^t R(s_t, a_t)]$.[1] A Bayes-optimal RL algorithm maximizes expected performance in MDPs sampled from prior $p(M)$, i.e., $\mathcal{J}(\bar{\pi}) = \mathbb{E}_{p(M)}[\mathcal{J}_M(\bar{\pi})]$ (Singh et al., 2009). The prior represents the domain, i.e., the distribution of MDPs that the agent will encounter (e.g., MiniGrid mazes (Chevalier-Boisvert et al., 2023) or Atari games). For clarity of exposition, we assume all MDPs possible under $p(M)$ share the same $\mathcal{S}, \mathcal{A}, \gamma$, so only $R, T, T_0$ are initially uncertain.[2] We use $p(M|h_t)$ to denote the posterior after updating $p(M)$ on the evidence in history $h_t$, i.e., $p(M|h_t) \propto p(h_t|M)p(M)$.

A BAMDP is thus defined by tuple $\bar{M} = (\bar{\mathcal{S}}, \mathcal{A}, \bar{R}, \bar{T}, \bar{T}_0, \gamma)$ where:

- $\bar{\mathcal{S}} = \mathcal{S} \times \mathcal{H}$, i.e., an information-augmented state space, so each state $\bar{s}_t = \langle s_t, h_t \rangle$.
- $\mathcal{A}$ and $\gamma$ are shared with the underlying MDPs, although internally $\bar{\pi}$ may sample its actions from an MDP policy $\pi_t$ that it learnt, i.e., $\bar{\pi}(a|\bar{s}_t) = \pi_t(a|s_t)$.
- $\bar{R}(\bar{s}_t, a) = \mathbb{E}_{p(M|h_t)}[R(s_t, a)]$, the expected reward under the current posterior.
- $\bar{T}(\bar{s}_{t+1}|\bar{s}_t, a_t) = \mathbb{E}_{p(M|h_t)}[T(s_{t+1}|s_t, a_t)R(r_{t+1}|s_t, a_t)\mathbb{1}[h_{t+1} = h_t a_t r_{t+1} s_{t+1}]]$
- $\bar{T}_0(\langle s_0, h_0 \rangle) = \mathbb{E}_{p(M)}[T_0(s)]\mathbb{1}[h_0 = s_0]$.

We illustrate the basic concepts of BAMDPs using the *caterpillar domain* shown in Fig. 2. Here, $p(M)$ represents how butterflies usually lay eggs on the best food source in the area, but 10% of the time a more rewarding source is nearby; upon hatching on the weed, the caterpillar does not know which MDP it is in and must decide whether exploring the neighboring bush is worth the energy and opportunity cost. The bush's reward varying across possible MDPs manifests as stochastic BAMDP dynamics at the transition where the caterpillar first observes it (e.g., taking the highlighted *eat* action). After observing the reward, $p(M|h_t)$ collapses to the underlying MDP and all dynamics become deterministic (e.g., at the transitions with red highlighted arrows).

---

[1] We can convert settings with episodic environments or train/test regimes to this form by augmenting the state space, e.g., by adding within-episode step indices or train/test indicators.

[2] This formulation can be extended to POMDPs and for distributions over $\mathcal{S}, \mathcal{A}, \gamma$ without any conceptual changes—the agent receives observations $o_t$, and expectations are taken over additional variables as needed.

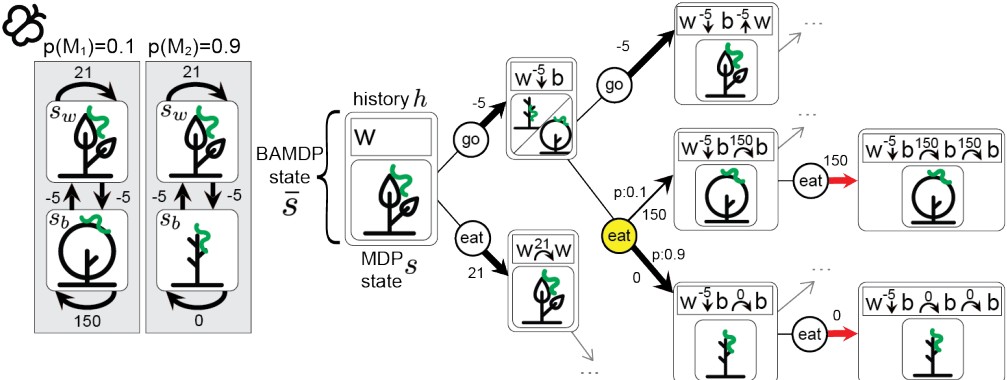

Figure 2: The caterpillar domain formulated as a BAMDP. Left: prior $p(M)$ is a categorical distribution over MDPs $M_1$ and $M_2$; in both, all transitions are deterministic, with the curved and straight arrows corresponding to *eat* and *go* actions respectively. The caterpillar hatches at state $s_w$, and must decide whether to *eat* for guaranteed reward 21, or incur $-5$ reward to *go* to $s_b$. Right: truncated BAMDP transition diagram, arrows are labeled with rewards (and transition probabilities if $p < 1$). The stochastic transitions (from the highlighted *eat* action) are due to the uncertainty over the MDP; all future transitions (the highlighted arrows) become deterministic once its identity is revealed.

The expected return of an RL algorithm in a BAMDP is equal to its expected return while learning in the initially unknown MDP $M \sim p(M)$:

$$
\begin{aligned}
\mathbb{E}_{\bar{\pi}}[\bar{G}] &= \mathbb{E}_{\bar{T}_0, \bar{T}} \left[ \sum_{t=0}^{\infty} \gamma^t \bar{R}(\bar{s}_t, \pi(\bar{s}_t)) \right] \\
&= \mathbb{E}_{p(M)} \left[ \mathbb{E}_M \left[ \sum_{t=0}^{\infty} \gamma^t R(s_t, \pi(\bar{s}_t)) \right] \right] = \mathbb{E}_{p(M)} \left[ \mathcal{J}_M(\bar{\pi}) \right] = \mathcal{J}(\bar{\pi}).
\end{aligned}
$$

Thus, an optimal policy for the BAMDP, $\bar{\pi}^*$ maximizing $\mathbb{E}_{\bar{\pi}}[\bar{G}]$, is a Bayes-optimal RL algorithm with respect to prior $p(M)$, i.e., it explores optimally for that domain.[3] Since $\mathbb{E}_{\bar{\pi}}[\bar{G}]$ is calculated through future information states, $\bar{\pi}^*$ must *plan through its own learning*. E.g., in Fig. 2 with large enough $\gamma$, $\bar{\pi}^*$ would deem $s_b$ worth exploring because it can stay there if it finds food, otherwise it will learn that $s_b$ is empty and use this knowledge to return to $s_w$ and never leave again.

## 3 PSEUDO-REWARDS CORRECT BAMDP VALUE MISESTIMATION

We first use our framework to explain how IM and reward shaping can incentivize effective exploration by signalling the value of BAMDP states.

### 3.1 THE RELATIONSHIP BETWEEN VALUE MISESTIMATION AND REGRET

RL algorithms typically act, implicitly or explicitly, to maximize a value prediction estimated from the history, which we denote by $\hat{\bar{Q}}(\bar{s}_t, a)$. Applying the performance difference lemma (Kakade & Langford, 2002) at the BAMDP level shows how algorithmic regret is directly related to how these estimates differ from the Bayes-Optimal value over their trajectory:

**Lemma 3.1.** *The Bayesian regret (Ghavamzadeh et al., 2015) of algorithms acting on estimate $\hat{\bar{Q}}(\bar{s}_t, a)$ can be expressed as:*

$$
\mathbb{E}_{\bar{\pi}^*}[\bar{G}] - \mathbb{E}_{\bar{\pi}}[\bar{G}] = \mathbb{E}_{\bar{\pi}} \left[ \sum_t \gamma^t \left( \bar{V}^*(\bar{s}_t) - \bar{Q}^* \left( \bar{s}_t, \arg\max_a \hat{\bar{Q}}(\bar{s}_t, a) \right) \right) \right], \tag{1}
$$

Thus, if we can add pseudo-rewards that nudge the value estimates of RL agents $\hat{\bar{Q}}$ such that they select actions with higher BAMDP value $\bar{Q}^*$, they will explore more optimally.

---

[3]Note that, since Bayes-optimal RL algorithms explore only when expected to increase return, they don't always explore enough to converge on exactly optimal MDP policies $\pi^*$. E.g., in the caterpillar domain, if $\gamma$ is small enough then immediate reward dominates $\mathbb{E}_{\bar{\pi}}[\bar{G}]$, so $\bar{\pi}^*$ eats at $s_w$ rather than risking delaying rewards by checking $s_b$ (see Appendix I.1 for full calculations).

## 3.2 BAMDP Value Decomposition

We can now understand the role of pseudo-rewards as directly incentivizing RL agents to go to more valuable BAMDP states. IM terms that reward novel observations or diverse actions encode the value of the information in the BAMDP state $h_t$; these are popular because commonly used RL algorithms do not inherently account for the value of information (see Appendix C). RL agents can also misestimate the value of physical states due to incorrect *initial* beliefs; many other pseudo-rewards compensate for this type of misestimated value, typically when there is significant prior knowledge about how to maximize rewards, e.g., it is advantageous for a ball-dribbling robot to be closer to the ball (Ji et al., 2023). This is helpful to express as a pseudo-reward because it can be difficult to program such prior knowledge into non-tabular (e.g., deep RL) algorithms before they begin learning. We decompose the BAMDP value function into these two values, which we call the Value of Information and Value of Opportunity, respectively.

**Definition 3.1** (Value of Information). The Value of Information (VOI) from state $\bar{s}_t$ is the increase in $\bar{\pi}^*$'s expected return from $s_t$ due to the information in $h_t$ compared to its initial beliefs:

$$\bar{V}_I^*(\langle s_t, h_t \rangle) = \bar{V}^*(\langle s_t, h_t \rangle) - \bar{V}^*(\langle s_t, h_0 \rangle). \tag{2}$$

E.g, $h_t$ after watching TV may contain more information than $h_0$, but $\bar{V}_I^*$ would be zero if that information can't help $\bar{\pi}^*$ get higher return. Meanwhile, exploring a new section of a maze, even if it had no rewards, lets $\bar{\pi}^*$ focus its search on a smaller remaining area, so $\bar{V}_I^*$ would be positive.

**Definition 3.2** (Value of Opportunity). The Value of Opportunity (VOO) to $\bar{\pi}$ from state $\bar{s}_t$ is the expected optimal value of state $s_t$ without having learnt anything, i.e.:

$$\bar{V}_O^*(\langle s_t, h_t \rangle) = \bar{V}^*(\langle s_t, h_0 \rangle). \tag{3}$$

E.g., if there is guaranteed reward at a known goal state $s_g$, then $s$ that are fewer steps from it have higher VOO. Conversely, if walking near an icy cliff edge always has a high chance of injury, then $s$ there have lower VOO.

**Lemma 3.2** (BAMDP Value Decomposition). *The optimal BAMDP value can be decomposed into the Value of Information and the Value of Opportunity:*

$$\bar{V}^*(\langle s_t, h_t \rangle) = \bar{V}_I^*(\langle s_t, h_t \rangle) + \bar{V}_O^*(\langle s_t, h_t \rangle). \tag{4}$$

We can now categorize IM and reward shaping terms by which of these components they signal (see Table 1). We can also understand the failure of pseudo-rewards to effectively guide RL algorithms as a result of them aligning poorly with the true BAMDP value. For example, negative surprise (Berseth et al., 2019) signals the prior knowledge that unpredictable parts of the environment have lower $\bar{V}_O^*$. This is well-aligned for environments with dangerous dynamics, but fails in safe environments where unpredictability correlates poorly with negative outcomes, causing agents to stare at a wall rather than exploring (Rhinehart et al., 2021). Meanwhile, entropy bonuses (Szepesvári, 2010; Mnih, 2016; Haarnoja et al., 2017) signal that more $\bar{V}_I^*$ can be gained by trying a wider spread of actions. This breaks down if the scale of the pseudo-rewards is too high, because overly random behavior is unlikely to reach interesting novel states. Thus, the bonuses must be carefully balanced with the scale and frequency of the extrinsic rewards (Hafner et al., 2023). See Appendix B for more in-depth discussion of these and more examples.

## 4 Preserving Optimality with BAMDP Potential-Based Shaping

Section 3 provides principles for designing pseudo-rewards to guide effective exploration towards states of increasing BAMDP value. Another important desideratum for pseudo-rewards is that they do not prevent RL agents from converging on good behaviors once exploration is complete. As our algorithms become more powerful, they may get better at finding "reward-hacking" behaviors that maximize shaped rewards to the detriment of the underlying rewards we care about. This can be avoided by using pseudo-rewards that *preserve optimality*, i.e., guarantee that any behavior maximizing shaped rewards also maximizes underlying rewards. Using our framework, we define a form of pseudo-reward which makes it easy to retrofit pseudo-reward functions, or design new terms, which preserve optimality: BAMDP Potential-based shaping Functions (BAMPFs). We prove that the use of BAMPFS can avoid reward-hacking in both RL (i.e., optimizing over MDP policies) and meta-RL (i.e., optimizing over RL algorithms) settings.

Table 1: Typology of IM and reward shaping terms based on the value components (Value of Information $\bar{V}_I^*$ and Value of Opportunity $\bar{V}_O^*$) they signal. Attractive signals increase with measures of underestimated value, and repulsive signals decrease with metrics warning of overestimated value.

|  | No $\bar{V}_O^*$ Signal | Repulsive $\bar{V}_O^*$ Signal | Attractive $\bar{V}_O^*$ Signal |
|---|---|---|---|
| **Attractive $\bar{V}_I^*$ Signal** | • Prediction error (58; 10) 
 • Count-based (6; 10) 
 • Entropy bonus (77; 53) 
 • Skill discovery (70; 81) 
 • Info gain (36; 71) | • Empowerment (43; 30) 
 • Information Capture (61) | • Optimism bonus (74; 60) 
 • Subtask unlocking (34) |
| **No $\bar{V}_I^*$ Signal** |  | • Surprise minimization (8) 
 • Information cost (22) 
 • Joint angle violation penalty (39) | • Goal proximity (56; 28) 
 • Subgoal reaching (59; 75) 
 • Ball possession (39) |

## 4.1 DEFINITION OF BAMDP POTENTIAL-BASED SHAPING FUNCTIONS

Classic potential-based shaping guides RL agents toward more valuable MDP states by encoding their desirability with the potential function $\phi(s)$ (e.g., Fig. 1a where $\phi(s)$ is the proximity to the goal state). A BAMDP Potential-Based Shaping Function (BAMPF) can guide RL agents by encoding the desirability of BAMDP states $\phi(\bar{s})$ (or equivalently, $\phi(h)$). This allows them to signal not only the MDP state value (i.e., VOO), but also the value of the information accumulated (VOI), encouraging exploration to gather valuable experiences.

**Definition 4.1.** (BAMPF) Let any $\bar{\mathcal{S}}, \mathcal{A}, \gamma$ be given. Function $F$ is a **BAMDP P**otential-Based Shaping **F**unction if there exists a real-valued function $\phi$ such that for all $h_t \in \mathcal{H} - \{h_0\}$,

$$F(h_t) = \gamma\phi(h_t) - \phi(h_{t-1}), \tag{5}$$

where $h_{t-1}$ is the first $t-1$ timesteps of $h_t$. See Fig. 1b for an example of a simple BAMPF based on the size of the set of MDP states visited in $h$.

E.g., *Information Gain* (Houthooft et al., 2016) expressed as the decrease in entropy of the agent's belief of the MDP dynamics $\hat{p}(T)$, i.e., $H(\hat{p}(T|h_{t-1})) - H(\hat{p}(T|h_t))$, can be viewed as a BAMPF for $\gamma = 1$ with $\phi(h) = -H(\hat{p}(T|h))$, i.e., the certainty of the posterior belief after updating on $h$, encouraging exploration towards states where the agent knows more about the environment. Due to the expressivity of $\phi$, it is straightforward to convert virtually any IM or reward shaping function to a BAMPF: take whatever measure of the desirability of the information and/or MDP state that it is based on, and use that measure as $\phi$.

## 4.2 BAMPFs PRESERVE OPTIMALITY IN META-RL

We first consider the use of BAMPFs for guiding meta-RL systems to discover better RL algorithms more efficiently, without misleading them to converge on suboptimal reward-hacking behaviors. In this setting, the meta-learner generates RL algorithms $\bar{\pi}_\theta$, updating $\theta$ to maximize $\bar{\pi}_\theta$'s expected performance in MDPs sampled from $p(M)$, with the goal of eventually meta-learning a Bayes-Optimal $\bar{\pi}^*$. Pseudo-rewards can be added to shape the returns the meta-RL agent observes, guiding it to generate better $\bar{\pi}_\theta$. E.g., entropy bonuses could encourage generation of $\bar{\pi}_\theta$ that explore a wider range of actions. But if the pseudo-reward doesn't preserve optimality, it could cause the meta-learner to converge on a $\bar{\pi}_\theta$ that explores badly with respect to the true task distribution. We extend PBSF theory to prove BAMPFs preserve optimality in BAMDPs, i.e., in meta-RL problems.

### 4.2.1 BAMDP POTENTIAL-BASED SHAPING THEOREM

We model the effect of pseudo-reward function $F(h_t)$ as producing *shaped BAMDP* $\bar{M}'$ with shaped reward $\bar{R}'(\bar{s}_t, a, \bar{s}_{t+1}) = \bar{R}(\bar{s}_t, a) + F(h_{t+1})$, while $\bar{\mathcal{S}}$ and $\bar{T}$ are unchanged, i.e., $h_t$ still only contains the underlying rewards. This reflects the fact that $F(h_t)$ is fully known from the start, so it

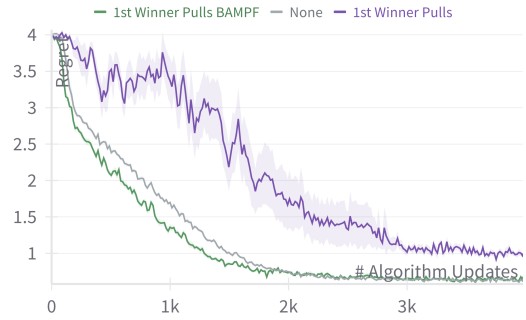 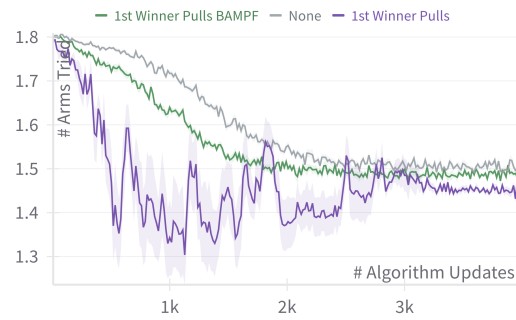

(a) Average regret of the learnt RL agents as they are updated through meta-learning.

(b) Average number of arms the learnt agents try in each MDP throughout meta-learning.

Figure 3: The effect of reward shaping on A2C meta-learning an RNN-based RL agent for Bernoulli Bandits with two arms, reward probabilities $(0.1, 0.9)$ and a budget of 10 pulls. The mean and standard error of 10 seeds are plotted for each condition. Without shaping, the meta-learner gradually learns to generate RL agents that try fewer arms on average, avoiding over-exploration (grey curve in 3b). The *1st Winner Pulls BAMPF* (green) sets $\phi$ to the pull count of the first arm that yielded a reward, helping A2C learn to exploit and achieve lower regret more quickly, while still converging to the optimal strategy. However, when this pull count is used directly as a pseudo-reward (*1st Winner Pulls*, purple), it causes the meta-learner to converge on an agent that over-exploits.

should not be treated as information or influence the posterior. The RL algorithm observes the underlying rewards in $h_t$, while only the meta-learner receives the pseudo-rewards in this setting. An optimal algorithm for $\bar{M}'$ maximizes the expected shaped return from $\bar{R}'$, i.e., $\bar{\pi}^{*\prime} = \arg\max_{\bar{\pi}} \mathbb{E}_{\bar{\pi}}[\bar{G}']$.

We now state our main theorem that BAMPFs always preserve Bayes-Optimality.

**Theorem 4.2** (BAMDP Potential-Based Shaping Theorem). *For a pseudo-reward function to guarantee that the optimal algorithm for any shaped BAMDP is optimal for the original BAMDP, i.e., Bayes-optimal for the underlying RL problem, it is necessary and sufficient for it to be a potential-based shaping function on the BAMDP state.*

*Proof Sketch.* For sufficiency, we show that the $\phi$ form a telescoping sum, reducing to a constant. For necessity, we construct a BAMDP such that when $F - (\gamma\phi' - \phi)$ is nonzero, different actions maximize the shaped and extrinsic returns. The proof follows Ng et al. (1999) but rewards impact the BAMDP state, making it more complex than for regular MDPs; see A.1 for full proofs. □

Thus, if the meta-learner converges to an RL algorithm $\bar{\pi}^{*\prime}$ that learns optimally for the BAMPF-shaped rewards, this algorithm will also always explore optimally with respect to the true reward distribution. Meanwhile, if we use a pseudo-reward function that is not a BAMPF, there will be settings where $\bar{\pi}^{*\prime}$ explores sub-optimally. We extend this result to prove that BAMPFS also preserve *approximate* optimality of RL algorithms in Appendix A.2.

### 4.2.2 EXPERIMENT: SHAPING META-RL ON BERNOULLI BANDITS

We demonstrate this result in the Bernoulli Bandit meta-RL problem introduced by Wang et al. (2016). Every MDP in $p(M)$ has two arms, one with reward probability 0.1 and the other 0.9 (randomly assigned), and a budget of 10 pulls. At each step an RNN-based RL agent observes the last arm that was pulled, its payout, and how many pulls it has left, and chooses which arm to pull next. After 10 pulls, the episode ends and a new MDP is sampled from $p(M)$. The meta-RL algorithm, A2C (Mnih, 2016), continually updates the RNN agent to maximize expected return in its 10-step lifetime, i.e., to find the RL strategy that optimally balances exploration and exploitation with respect to $p(M)$. Fig. 3b shows how without shaping (the grey curve), the meta-learner gradually learns to generate RL agents that try fewer arms on average, i.e., avoiding over-exploration. This shows that the meta-learner initially over-estimates the VOI of trying more arms, so we could add a BAMPF to correct its prior. The *1st Winner Pulls BAMPF* sets $\phi(h)$ to the total pull count of the first arm that produced a reward, decreasing the relative perceived VOI of exploring the other arm. Fig. 3b (green curve) shows it helps A2C learn more exploitative RL agents more quickly, and

Fig. 3a shows that these agents achieve lower regret more quickly, while it doesn't prevent eventual convergence to an optimal agent. In contrast, when this pull count is used directly as a pseudo-reward (purple curve), it causes the meta-learner to converge on an agent that exploits too heavily and never achieves the optimal regret, because whenever the $p = 0.1$ arm is by chance the first winner, the shaped return is maximized by continuing to pull it, regardless of its future payouts. See Appendix G for full experimental details.

## 4.3 Preserving Optimality in RL with BAMPFs

Now we turn our attention to the common RL setting, where an RL algorithm learns an MDP policy to maximize return. A certain natural type of BAMPF also resists hacking at the MDP level. These are BAMPFS based on $\phi$ that are bounded monotone increasing functions over time, i.e., $\phi(h_{t_1}) \leq \phi(h_{t_2})$ for all realizable histories at training steps $t_1 < t_2$ (so $h_{t_1}$ is a prefix of $h_{t_2}$).

**Theorem 4.3.** *If the pseudo-rewards added to an MDP can be expressed as a BAMPF with a bounded potential that is monotone increasing over time, it will eventually preserve approximate optimality in the MDP, i.e.,*

$$\forall \epsilon > 0 \quad \exists H : \forall t > H : \mathbb{E}_{\pi_t^{*'}}[G] > \mathbb{E}_{\pi^*}[G] - \epsilon, \tag{6}$$

*where $t$ is the training step, $\mathbb{E}_\pi[G]$ denotes expected unshaped return of an MDP policy $\pi$, $\pi_t^{*'}$ is a policy maximizing the shaped return at step $t$, and $\pi^*$ is an optimal policy for the underlying MDP.*

*Proof Sketch.* The BAMPF rewards form a telescoping sum over any MDP episode, leaving a single policy-dependent term: $\phi$ at the final step of the episode. Because $\phi$ is bounded and monotone, there must be a final step in training $H$ where it increases by at least $\epsilon$, after which the artificial advantage it can give the chosen policy is less than $\epsilon$. Therefore the optimal policy in the shaped MDP must be less than $\epsilon$ worse than the optimal policy in the true MDP. See Appendix A.3 for the full proof. □

This class of BAMPF naturally includes intrinsic motivation terms that signal the total value of information gathered so far, e.g., count-based rewards (Bellemare et al., 2016) or information gain (Houthooft et al., 2016), and they are simple and intuitive to design. For example, take a single-state MDP with 100 levers only one of which yields a reward when pulled. A bounded monotone $\phi$ could be the number of unique levers tried so far, encoding the VOI from having explored more levers. Fig. 4a shows how, when applied to DQN, the BAMPF reward converges to a constant as more levers are tried. Fig. 4b shows how this BAMPF causes the agent to try more levers on average than without pseudo-rewards (which relies purely on $\epsilon-$greedy exploration). And yet, it doesn't prevent DQN from converging to the optimal policy, i.e., pulling the correct lever at every step (Fig. 4c). Meanwhile, an entropy bonus (the entropy of the last 10 lever pulls, which isn't a BAMPF) makes the agent oscillate between pulling the correct lever to get the extrinsic reward, and pulling other levers to increase the entropy bonus, which achieves higher shaped rewards but lower true rewards than pulling the correct lever at every step.

### 4.3.1 Experiment: Shaping RL in Mountain Car

We now demonstrate the benefits of BAMPFs in a more realistic setting: with PPO (Schulman et al., 2017) in the Mountain Car environment (Brockman et al., 2016) which has a continuous state comprising the car's position and velocity. The reward is -1 at each timestep; each episode lasts 200 steps or ends early if the car reaches the goal. The car starts in a valley, and to reach the goal (the flagpole in Fig. 5a) it must build momentum by first moving up the *opposite* slope. Thus, displacement from the lowest point in the valley is an intuitive shaping reward to signal the VOO of being further uphill in either direction. We find this helps PPO reach greater displacements early in training (red curve in Fig. 5c), but eventually results in reward-hacking policies that collect more pseudo-rewards by avoiding the goal (Figures 5b and 9). Converting this to classic potential-based shaping by setting $\phi(s)$ to the displacement, we find it preserves optimality but doesn't help learning (blue curve in Fig. 5b), possibly because the further uphill the car gets, the more negative PBS rewards it soon receives when it rolls back. We can understand these failures as a result of the fact that displacement is too weak a signal for the true VOO in Mountain Car. Instead, we could signal VOI by rewarding the *maximum* displacement so far, which doesn't penalize temporary decreases in displacement. This depends on more than the current state, so it isn't a valid MDP potential

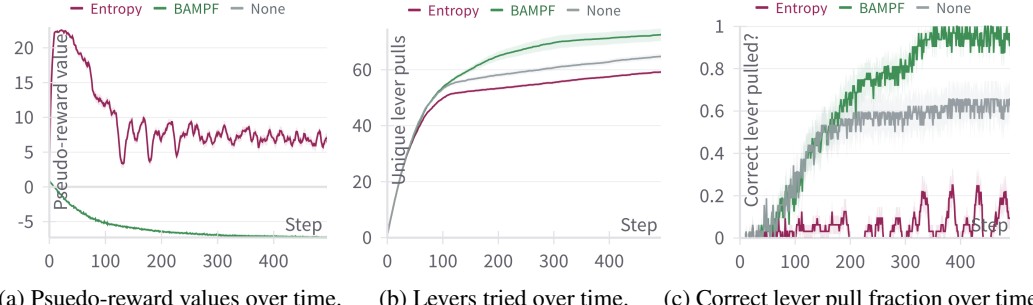

(a) Psuedo-reward values over time.  (b) Levers tried over time.  (c) Correct lever pull fraction over time.

Figure 4: The effect of a bounded monotone BAMPF and entropy bonus pseudo-rewards on DQN in a 1-state MDP, where 1 in 100 total levers gives reward 10 when pulled. *None* refers to DQN without any pseudo-rewards. The *BAMPF* potential is the count of unique levers tried, and the *Entropy* reward is 10x the entropy of the last 10 lever pulls. The setting is non-episodic with $\gamma = 0.9$. The mean and standard error of 32 seeds are plotted for each condition. See Appendix F for full details.

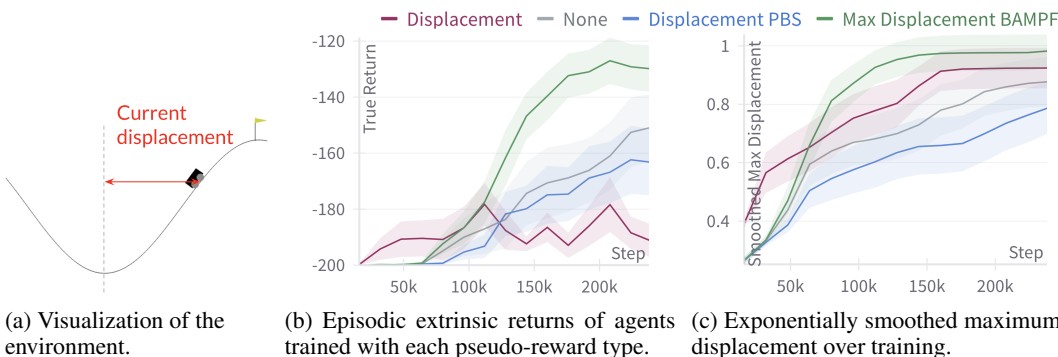

(a) Visualization of the environment.  (b) Episodic extrinsic returns of agents trained with each pseudo-reward type.  (c) Exponentially smoothed maximum displacement over training.

Figure 5: The effect of pseudo-rewards on PPO in *Mountain Car*; the mean and standard error of 10 seeds are plotted for each type of reward shaping. *Displacement* rewards the current $x$ displacement of the car, *Displacement PBS* is potential-based shaping with the current displacement as the MDP state potential $\phi(s)$, and *Max Displacement BAMPF* uses the exponentially smoothed maximum displacement over training (5c) as the BAMDP state potential $\phi(h)$. With *Displacement* the agent learns a reward-hacking policy that avoids the goal to collect more pseudo-rewards (see Fig. 9), while the BAMPF helps PPO learn to reach the goal more quickly while preserving optimality (5b).

but it *is* a valid BAMDP potential $\phi(h_t)$. It is bounded due to the finite width of the environment, and as the maximum it monotonically increases throughout training. It signals the VOI of the RL agent learning how to get further uphill; because it sometimes surpasses its maximum displacement by chance before learning how to consistently get that far, we apply exponential smoothing. Fig. 5c (green curve) shows the resulting BAMPF gets the agent to explore further more quickly, and Fig. 5b shows it learns to reach the goal and maximize true rewards sooner while avoiding reward-hacking. See Appendix H for full experimental details.

### 4.3.2 CONVERTING PSEUDO-REWARDS TO RESIST REWARD-HACKING IN RL

Although virtually any pseudo-reward can be cast as a BAMPF, the monotonicity condition doesn't naturally include non-cumulative functions of just the latest behavior, e.g., prediction error (the predictability of the latest observations) or entropy bonuses (the entropy of the latest policy) which can fluctuate up or down. However, these terms would still preserve optimality if they could be converted to the sum of a bounded monotone increasing BAMPF and a classic PBSF. E.g., Curiosity (Pathak et al., 2017) rewards the error of a learnt dynamics model in predicting the latest observations, to encourage continually seeking novel experiences while improving the dynamics model. The non-monotone component that encourages visiting novel states could be captured by a PBSF with $\phi(s)$ equal to the error of a *fixed model*'s prediction of features of $s$, similar to RND (Burda et al., 2018). The value of improving the dynamics model can be captured by a BAMPF with $\phi(h)$ equal to the *maximum* accuracy so far of the dynamics model on a *fixed set of transitions*. To prevent stagnation,

the fixed model and transition set could be refreshed regularly between batch updates– as long as they can't change *within* an episode, optimality will be preserved. This form of Curiosity would not be susceptible to the Noisy TV Problem, since the dynamics model's accuracy $\phi(h)$ could not increase indefinitely while watching TV, and even if the fixed model's error $\phi(s)$ were higher at the TV, Ng et al. (1999)'s result guarantees that a policy can't maximize total shaped return by staying there. See Appendix D for a more detailed exposition and numerical demonstration.

## 5 RELATED WORK

**Theory of Intrinsic Motivation** Oudeyer & Kaplan (2007) categorize IM as knowledge-based, competence-based or morphological, and evaluate them by their exploration (leading to exploratory and investigative behavior) and organization (leading to structured and organized behavior) potentials, which are similar to the effects of signalling $\bar{V}_O$ and $\bar{V}_I$. Singh et al. (2009) propose an evolutionary framework for rewards as maximizing expected fitness over a distribution of environments, concluding that optimal reward depends on regularities across the distribution and properties of the learning agent. Aubret et al. (2023) propose an information-theoretic taxonomy of IM, but this only captures IM terms signalling value of information; they don't consider the prior environment distribution and so cannot explain how pseudo-rewards should be designed for a given domain.

**Extensions of Potential-Based Reward Shaping** Devlin & Kudenko (2012) and Forbes et al. (2024) extend PBS theory to potential functions that vary over time, including functions of the history, showing that they also preserve the optimal MDP policy. However, Devlin & Kudenko (2012)'s proof omits the episodic setting, and Forbes et al. (2024) modify the potential at the last step of each episode to cancel out all prior pseudo-rewards. This is inappropriate for potentials measuring the value of information, since observations from an episode do not lose all value once the episode ends. Eck et al. (2016) extend PBSFs to POMDP planning, defining potential functions over POMDP belief states and categorizing them as Domain-dependent and Domain-independent, which share similarities to $\bar{V}_O$ and $\bar{V}_I$. Kim et al. (2015) propose BAMDP potential-based shaping specifically for a model-based Bayesian RL method, although they do not prove that the PBS theorem still holds for BAMDPs (which is non-trivial, because rewards influence the BAMDP state) and do not consider the broader implications for developing pseudo-rewards for any RL algorithm.

See Appendix E for an extended discussion of related work.

## 6 DISCUSSION

By formulating RL problems as BAMDPs, we formalize how intrinsic motivation and reward shaping help exploration by signalling the value of BAMDP states, which we decompose into the value of information and the value of opportunity. This allows us to characterize the roles of existing pseudo-rewards, explains when they impede exploration as misalignment with these values, and provides principles for designing new terms to guide learning in any given domain. This framework also gives us tools to tackle reward-hacking, i.e., to avoid incentivizing RL agents to converge on suboptimal final behaviors. We extend potential-based shaping theory to propose BAMPFs, a form of pseudo-reward that can avoid reward-hacking both in the typical RL setting, by preserving the optimality of the learnt MDP policies, and in meta-RL, by preserving the optimality of the learnt RL algorithms. Although it is a manual process, it is straightforward to design and convert many existing pseudo-rewards into this form, as we illustrate with a case-study converting a prediction error IM term to avert the Noisy TV problem. Our experiments in the Mountain Car RL environment and Bernoulli Bandits meta-RL domain demonstrate the utility of our framework for designing practical pseudo-reward functions which both improve learning efficiency and avoid reward-hacking.

## 7 ACKNOWLEDGMENTS

We would like to thank Cameron Allen, Justin Svegliato and Christian Guckelsberger for insightful discussions, and Benjamin Plaut, Micah Carroll, Niklas Lauffer, Hanlin Zhu and David Abel for feedback on drafts. We are also grateful to Cameron and Niklas for implementation advice. This material is supported in part by the Center for Human-Compatible AI (CHAI) and the Fannie and John Hertz Foundation.

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

## A  POTENTIAL-BASED SHAPING PROOFS

### A.1  MAIN THEOREM

**Theorem 4.2** (BAMDP Potential-Based Shaping Theorem)**.** *For a pseudo-reward function to guarantee that the optimal algorithm for any shaped BAMDP is optimal for the original BAMDP, i.e., Bayes-optimal for the underlying RL problem, it is necessary and sufficient for it to be a potential-based shaping function on the BAMDP state.*

*Proof.* (Sufficiency) Denote the original BAMDP $\bar{M} = (\bar{\mathcal{S}}, \mathcal{A}, \bar{R}, \bar{T}, \bar{T}_0, \gamma)$ with optimal algorithm $\bar{\pi}^*$, and the shaped BAMDP as $\bar{M}' = (\bar{\mathcal{S}}, \mathcal{A}, \bar{R}', \bar{T}, \bar{T}_0, \gamma)$, i.e., it is identical to $\bar{M}$ except for its shaped reward function $\bar{R}'(\bar{s}_t, a, \bar{s}_{t+1}) = \bar{R}(\bar{s}_t, a) + F(\bar{s}_{t+1}) = \bar{R}(\bar{s}_t, a) + \gamma\phi(h_{t+1}) - \phi(h_t)$. So we model the pseudo-rewards as being added to the RL algorithm's reward signal internally, rather than entering through the history in the BAMDP state $\bar{s}_t$. We denote the optimal algorithm for $\bar{M}'$ by $\bar{\pi}^{*\prime}$.

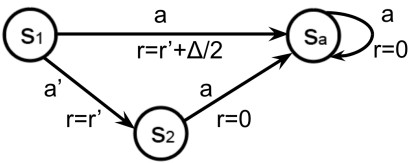

Figure 6: All edges have probability 1, and only action $a$ is applicable from $s_2$ and $s_a$.

Abbreviating $\phi(h_t)$ to $\phi_t$, we can now express the expected return of an algorithm in $\bar{M}'$ in terms of the underlying return it would achieve in $\bar{M}$:

$$
\begin{aligned}
\mathbb{E}_{\bar{\pi}}[\bar{G}'] &= \mathbb{E}_{\bar{T},\bar{T}_0,\bar{\pi}}\left[\sum_t \gamma^t \bar{R}'(\bar{s}_t, a_t, \bar{s}_{t+1})\right] \\
&= \mathbb{E}_{\bar{T},\bar{T}_0,\bar{\pi}}\left[\sum_t \gamma^t \left(\bar{R}(\bar{s}_t, a_t) + \gamma\phi_{t+1} - \phi_t\right)\right] \\
&= \mathbb{E}_{\bar{T},\bar{T}_0,\bar{\pi}}\left[\sum_{t=0}^{\infty} \gamma^t \bar{R}(\bar{s}_t, a_t)\right] + \mathbb{E}_{\bar{T},\bar{T}_0,\bar{\pi}}\left[-\phi_0 + \sum_{t=1}^{\infty} \gamma^t(\phi_t - \phi_t)\right] \\
&= \mathbb{E}_{\bar{\pi}}[\bar{G}] - \phi_0
\end{aligned}
\tag{7}
$$

Plugging this into the definitions of the Bayes-optimal algorithms for $\bar{M}'$ and $\bar{M}$:

$$
\begin{aligned}
\bar{\pi}^{*\prime} &\in \arg\max_{\bar{\pi}} \mathbb{E}_{\bar{\pi}}[\bar{G}'] \\
&= \arg\max_{\bar{\pi}} \mathbb{E}_{\bar{\pi}}[\bar{G}] - \phi_0 \\
&= \arg\max_{\bar{\pi}} \mathbb{E}_{\bar{\pi}}[\bar{G}] = \bar{\pi}^*
\end{aligned}
\tag{8}
$$

$\square$

We now prove that if pseudo-reward function $F(h)$ is not a BAMPF, then there exists BAMDP reward and transition functions such that no Bayes-optimal algorithm for the shaped BAMDP is Bayes-optimal in the original BAMDP.

*Proof.* (Necessity) Our proof follows Ng et al. (1999)'s proof of necessity, but is more complex because the BAMDP state contains the entire history. Assume $F$ is not a BAMPF. We need to show that we can construct $\bar{T}, \bar{R}$ such that no optimal algorithm in shaped BAMDP $\bar{M}'$ is also optimal in $\bar{M}$. Define $\phi(h) = -\sum_{t=0}^{\infty} \gamma^t F(h(a0s_a)_{t+1})$ for all $h$ with $s_a \in \mathcal{S}, a \in \mathcal{A}$, where $a0s_a$ denotes the trajectory segment from taking action $a$, receiving reward $0$ and transitioning to state $s_a$, $(a0s_a)_{t+1}$ denotes the trajectory formed by repeating this segment $t+1$ times, and $h(a0s_a)_{t+1}$ is the history formed by concatenating $h$ with $(a0s_a)_{t+1}$. By assumption of $F$ not being a BAMPF, there exists a valid transition between two BAMDP states $\langle s_1, h_1 \rangle, \langle s_2, h_1 a'r's_2 \rangle \in \bar{\mathcal{S}}$ via some action $a' \in \mathcal{A}$ yielding reward $r'$ such that $\gamma\phi(h_1 a'r's_2) - \phi(h_1) \neq F(h_1 a'r's_2)$. Define $\Delta = F(h_1 a'r's_2) - \gamma\phi(h_1 a'r's_2) + \phi(h_1)$. We now construct $\bar{M}$ with no uncertainty over $R, T$, i.e., $p(M)$ is concentrated on a single MDP $M$, illustrated in Fig. 6. In this $M$, we have $T(s_a|s_1, a) = T(s_2|s_1, a') = 1.0$, and from $s_2$ and $s_a$ the only applicable action $a$ leads to $s_a$ with probability 1. The initial MDP state is $s_1$, so for $s_1, h_1$ to be realizable in this environment $h_1$ is the length 1 sequence $s_1$. Let $R(s_1, a) = r' + \Delta/2, R(s_1, a') = r', R(\cdot, \cdot) = 0$ elsewhere. Then we have

$$\bar{Q}^*(\langle s_1, h_1 \rangle, a) = r' + \Delta/2$$

$$\bar{Q}^*(\langle s_1, h_1 \rangle, a') = r'$$

$$\bar{Q}^{*\prime}(\langle s_1, h_1 \rangle, a) = r' + \Delta/2 + \sum_{t=0}^{\infty} \gamma^t F(h_1(a0s_a)_{t+1})$$

$$= r' + \Delta/2 - \phi(h_1)$$

$$= r' + \Delta/2 + F(h_1 a' r' s_2) - \gamma\phi(h_1 a' r' s_2) - \Delta$$

$$= r' + F(h_1 a' r' s_2) - \gamma\phi(h_1 a' r' s_2) - \Delta/2$$

$$\bar{Q}^{*\prime}(\langle s_1, h_1 \rangle, a') = r' + F(h_1 a' r' s_2) + \gamma\sum_{t=0}^{\infty} \gamma^t F(h_1 a' r' s_2(a0s_a)_{t+1})$$

$$= r' + F(h_1 a' r' s_2) - \gamma\phi(h_1 a' r' s_2).$$

Hence

$$\bar{\pi}^*(\langle s_1, h_1 \rangle) = \begin{cases} a & \text{if } \Delta/2 > 0 \\ a' & \text{otherwise} \end{cases} \qquad \bar{\pi}^{*\prime}(\langle s_1, h_1 \rangle) = \begin{cases} a' & \text{if } \Delta/2 > 0 \\ a & \text{otherwise} \end{cases} \tag{9}$$

$\square$

## A.2   Near-Optimal Algorithms are Nearly Invariant to BAMPFs

We now extend our results to approximately optimal algorithms, first providing an overview of the results and proof sketches before diving into the full proofs.

### A.2.1   Approximate Optimality Corollaries

**Corollary A.1.** *With BAMPF shaping, a near-optimal algorithm $\bar{\pi}^\epsilon$ for $\bar{M}'$ will also be near-optimal when applied to $\bar{M}$, i.e.,*

$$\mathbb{E}_{\bar{\pi}^{*\prime}}[\bar{G}'] - \mathbb{E}_{\bar{\pi}}^\epsilon[\bar{G}'] < \epsilon \iff \mathbb{E}_{\bar{\pi}^*}[\bar{G}] - \mathbb{E}_{\bar{\pi}}^\epsilon[\bar{G}] < \epsilon. \tag{10}$$

*Proof Sketch.* Because BAMPF shaping shifts BAMDP returns by a constant, lower shaped return $\mathbb{E}_{\bar{\pi}}[\bar{G}']$ corresponds directly to $\mathbb{E}_{\bar{\pi}}[\bar{G}]$ decreasing by the same amount. See A.2.2 for the full proof.

$\square$

Another form of approximate optimality is the maximization of return over a finite horizon $k$. We can also upper-bound the regret as a function of $k$.

**Definition A.1** (k-step optimal). We define an RL algorithm as k-step optimal for a BAMDP if it maximizes the expected k-step return:

$$\bar{\pi}_k^* \in \arg\max_{\bar{\pi}} \mathbb{E}_{\bar{T}_0, \bar{T}} \left[ \sum_{t=0}^k \gamma^t \bar{R}(\bar{s}_t, \pi(\bar{s}_t)) \right]. \tag{11}$$

**Corollary A.2.** *If $F$ is a BAMPF with potential function of maximum magnitude $\phi_{\max}$, and the extrinsic reward has maximum magnitude $R_{\max}$, then the k-step optimal algorithm for the shaped BAMDP, $\bar{\pi}_k^{*\prime}$, has regret bounded by $2\gamma^k(\phi_{\max} + R_{\max}\frac{\gamma}{1-\gamma})$ in the underlying BAMDP, i.e.,*

$$\mathbb{E}_{\bar{\pi}^*}[\bar{G}] - \mathbb{E}_{\bar{\pi}_k^{*\prime}}[\bar{G}] \leq 2\gamma^k \left( \phi_{\max} + R_{\max}\frac{\gamma}{1-\gamma} \right). \tag{12}$$

*Proof Sketch.* The telescoping $\phi$ summed over horizon $k$ leaves a trailing term of $\gamma^k \phi_k$ from the last step, which allows us to bound $\bar{\pi}_k^{*\prime}$'s k-step regret compared to $\bar{\pi}_k^*$ in terms of $\phi_{\max}$. The regret of $\bar{\pi}_k^*$ compared to the fully Bayes-optimal algorithm $\bar{\pi}^*$ is bounded by the worst-case regret after step $k$, giving us a bound in terms of $R_{\max}$. See A.2.2 for the full proof.

$\square$

A.2.2 FULL PROOFS

The result in Equation 7 tells us that a near-optimal algorithm for $\bar{M}'$ will also be near-optimal in $\bar{M}$ because the return is just shifted by a constant.

**Corollary A.1.** *With BAMPF shaping, a near-optimal algorithm $\bar{\pi}^\epsilon$ for $\bar{M}'$ will also be near-optimal when applied to $\bar{M}$, i.e.,*

$$\mathbb{E}_{\bar{\pi}^{*\prime}}[\bar{G}'] - \mathbb{E}_{\bar{\pi}}^\epsilon[\bar{G}'] < \epsilon \iff \mathbb{E}_{\bar{\pi}^*}[\bar{G}] - \mathbb{E}_{\bar{\pi}}^\epsilon[\bar{G}] < \epsilon. \tag{10}$$

*Proof.*

$$\begin{aligned}
\mathbb{E}_{\bar{\pi}^{*\prime}}[\bar{G}'] - \mathbb{E}_{\bar{\pi}}[\bar{G}'] &= \mathbb{E}_{\bar{\pi}^{*\prime}}[\bar{G}] - \phi_0 - (\mathbb{E}_{\bar{\pi}}[\bar{G}] - \phi_0) \\
&= \mathbb{E}_{\bar{\pi}^{*\prime}}[\bar{G}] - \mathbb{E}_{\bar{\pi}}[\bar{G}] \\
&= \mathbb{E}_{\bar{\pi}^*}[\bar{G}] - \mathbb{E}_{\bar{\pi}}[\bar{G}]
\end{aligned} \tag{13}$$

$\square$

Now to prove Corollary A.2 we first introduce the following Lemma:

**Lemma A.1.** *If $F$ is a BAMPF with potential function of maximum magnitude $\phi_{\max}$, then the k-step optimal algorithm for the shaped BAMDP, $\bar{\pi}_k^{*\prime}$, has k-step regret bounded by $2\gamma^k \phi_{\max}$ in the true BAMDP:*

$$\max_{s,h} |\phi(h)| = \phi_{\max} \implies \mathbb{E}_{\bar{\pi}_k^*}[\bar{G}_k] - \mathbb{E}_{\bar{\pi}_k^{*\prime}}[\bar{G}_k] \le 2\gamma^k \phi_{\max} \tag{14}$$

*Proof.* First, observe that when PBSF rewards are summed over a finite horizon, all but the potentials of the first and last timesteps cancel out:

$$\begin{aligned}
\sum_{t=0}^{k} \gamma^t (\gamma \phi_{t+1} - \phi_t) &= -\phi_0 + \sum_{t=1}^{k-1} \gamma^t (\phi_t - \phi_t) + \gamma^k \phi_k \\
&= \gamma^k \phi_k - \phi_0
\end{aligned} \tag{15}$$

Thus, following the same steps as Equation 7 we can express the k-step return of an algorithm in the shaped BAMDP as:

$$\mathbb{E}_{\bar{\pi}}[\bar{G}'_k] = \mathbb{E}_{\bar{\pi}}[\bar{G}_k] + \mathbb{E}_{\bar{T},\bar{\pi}}[\gamma^k \phi_k] - \phi_0. \tag{16}$$

And so the k-step optimal algorithm in the shaped BAMDP can be expressed as:

$$\bar{\pi}_k^{*\prime} = \arg\max_{\bar{\pi}} \mathbb{E}_{\bar{\pi}}[\bar{G}'_k] = \arg\max_{\bar{\pi}} \mathbb{E}_{\bar{\pi}}[\bar{G}_k] + \mathbb{E}_{\bar{T},\bar{\pi}}[\gamma^k \phi_k] \tag{17}$$

Evaluating this expression at $\bar{\pi}_k^{*\prime}$ and applying the bound on the potential function, we get:

$$\begin{aligned}
\mathbb{E}_{\bar{\pi}_k^{*\prime}}[\bar{G}_k] + \mathbb{E}_{\bar{T},\bar{\pi}_k^{*\prime}}[\gamma^k \phi_k] &= \max_{\bar{\pi}} \mathbb{E}_{\bar{\pi}}[\bar{G}_k] + \mathbb{E}_{\bar{T},\bar{\pi}}[\gamma^k \phi_k] \\
&\ge \max_{\bar{\pi}} \mathbb{E}_{\bar{\pi}}[\bar{G}_k] - \gamma^k \phi_{\max} \\
&= \mathbb{E}_{\bar{\pi}_k^*}[\bar{G}_k] - \gamma^k \phi_{\max}
\end{aligned} \tag{18}$$

Rearranging the above inequality and applying the bound once more we get:

$$\begin{aligned}
\mathbb{E}_{\bar{\pi}_k^*}[\bar{G}_k] - \mathbb{E}_{\bar{\pi}_k^{*\prime}}[\bar{G}_k] &\le \gamma^k \phi_{\max} + \mathbb{E}_{\bar{T},\bar{\pi}_k^{*\prime}}[\gamma^k \phi_k] \\
&\le 2\gamma^k \phi_{\max}.
\end{aligned} \tag{19}$$

$\square$

**Corollary A.2.** *If $F$ is a BAMPF with potential function of maximum magnitude $\phi_{\max}$, and the extrinsic reward has maximum magnitude $R_{\max}$, then the k-step optimal algorithm for the shaped BAMDP, $\bar{\pi}_k^{*\prime}$, has regret bounded by $2\gamma^k(\phi_{\max} + R_{\max}\frac{\gamma}{1-\gamma})$ in the underlying BAMDP, i.e.,*

$$\mathbb{E}_{\bar{\pi}^*}[\bar{G}] - \mathbb{E}_{\bar{\pi}_k^{*\prime}}[\bar{G}] \leq 2\gamma^k \left(\phi_{\max} + R_{\max}\frac{\gamma}{1-\gamma}\right). \tag{12}$$

*Proof.* Since magnitude of the reward is bounded by $R_{\max}$, we can upper bound the expected return of $\bar{\pi}^*$ by the optimal k-step return $\mathbb{E}_{\bar{\pi}_k^*}[\bar{G}_k]$ plus the maximum return from step $k+1$ onwards:

$$\begin{aligned} \mathbb{E}_{\bar{\pi}^*}[\bar{G}] &\leq \mathbb{E}_{\bar{\pi}_k^*}[\bar{G}_k] + \gamma^{k+1}\frac{1}{1-\gamma}R_{\max} \\ &\leq \mathbb{E}_{\bar{\pi}_k^{*\prime}}[\bar{G}_k] + 2\gamma^k\phi_{\max} + \gamma^{k+1}\frac{1}{1-\gamma}R_{\max}, \end{aligned} \tag{20}$$

where we applied Lemma A.1 to substitute $\mathbb{E}_{\bar{\pi}_k^*}[\bar{G}_k]$ at the second line. Meanwhile, the full expected return of $\bar{\pi}_k^{*\prime}$ can be lower bounded by its k-step return plus the minimum return from step $k+1$ onwards:

$$\begin{aligned} \mathbb{E}_{\bar{\pi}_k^{*\prime}}[\bar{G}] &\geq \mathbb{E}_{\bar{\pi}_k^{*\prime}}[\bar{G}_k] - \gamma^{k+1}\frac{1}{1-\gamma}R_{\max} \\ &\geq \mathbb{E}_{\bar{\pi}^*}[\bar{G}] - 2\gamma^k\phi_{\max} - 2\gamma^{k+1}\frac{1}{1-\gamma}R_{\max}, \end{aligned} \tag{21}$$

where we substituted the $\mathbb{E}_{\bar{\pi}_k^{*\prime}}[\bar{G}_k]$ in the first line using the result in Equation 20. $\square$

*Remark.* For RL algorithms that have a minimum resolution at which they distinguish returns, BAMPF shaping ceases to affect their behavior once they are optimal over a long enough horizon. More precisely, we call an RL Algorithm that does not distinguish returns less than $d$ apart $d$-insensitive. For k high enough that $2\gamma^k\phi_{\max} < d$, all k-step optimal $d$-insensitive algorithms in a BAMPF-shaped BAMDP are behaviorally equivalent to their counterparts in the original BAMDP.

## A.3 Bounded Monotone Increasing Potentials Preserve MDP Optimality

**Theorem 4.3.** *If the pseudo-rewards added to an MDP can be expressed as a BAMPF with a bounded potential that is monotone increasing over time, it will eventually preserve approximate optimality in the MDP, i.e.,*

$$\forall\epsilon > 0 \quad \exists H : \forall t > H : \mathbb{E}_{\pi_t^{*\prime}}[G] > \mathbb{E}_{\pi^*}[G] - \epsilon, \tag{6}$$

*where $t$ is the training step, $\mathbb{E}_\pi[G]$ denotes expected unshaped return of an MDP policy $\pi$, $\pi_t^{*\prime}$ is a policy maximizing the shaped return at step $t$, and $\pi^*$ is an optimal policy for the underlying MDP.*

*Proof.* We first introduce a lemma to bound the maximum increase in the potential function:

**Lemma A.2.** *If $\phi$ is bounded and monotone, then for any RL algorithm $\bar{\pi}$ and finite $\delta > 0$ we have some point in time $H$ at which $\phi$ has changed by $\delta$ or more for the last time, and thus all future values of $\phi(h_t)$ must be within $\delta$ of each other, i.e.:*

$$\forall\bar{\pi}, \delta > 0, \Delta t \geq 0 \quad \exists H : |\phi(h_{t+\Delta t}) - \phi(h_t)| < \delta \quad \forall t > H. \tag{22}$$

*Proof.* Because the sequence of $\phi(h_t)$ is bounded and monotone, the Monotone Convergence Theorem tells us that it is convergent. For all convergent sequences $S_t$,

$$\forall\delta > 0 \quad \exists H : |S_\infty - S_t| < \delta \quad \forall t > H, \tag{23}$$

where $S_\infty$ is the infimum or supremum of the sequence. Finally, because the sequence is monotone we have $|S_{t+\Delta t} - S_t| \leq |S_\infty - S_t|$, therefore $|S_{t+\Delta t} - S_t| < \delta$. $\square$

We will use this lemma to bound the artificial advantage any policy output by the RL algorithm can get due to shaping, and thus show that any policy chosen to maximize shaped return is still near-optimal for the real return.

We proceed by expressing the expected return of policy $\pi_t$ output by $\bar{\pi}$ and shaped by the BAMPF at training step $t$, $\mathbb{E}_{\pi_t}[G'_t]$ (i.e., the expected discounted sum of both pseudo-rewards and extrinsic rewards that $\pi_t$ achieves in one episode; $G'_t$ also varies with $t$ because the BAMPF varies over time), in terms of the unshaped return $\mathbb{E}_{\pi_t}[G]$ (i.e., the discounted sum of extrinsic rewards that the same policy $\pi_t$ achieves in one episode) and the episode length $N$:

$$\mathbb{E}_{\pi_t}[G'_t] = \mathbb{E}_{\pi_t}[G + \gamma\phi(h_{t+1}) - \phi(h_t) + \gamma^2\phi(h_{t+2}) - \gamma\phi(h_{t+1})... + \gamma^N\phi(h_{t+N})] \quad (24)$$
$$= \mathbb{E}_{\pi_t}[G] - \phi(h_t) + \gamma^N\mathbb{E}_{\pi_t}[\phi(h_{t+N})]$$

We use this to express the difference in shaped return between MDP policy $\pi_t$ that was output by the RL algorithm at step $t$, and any other MDP policy $\pi$, if we rolled them out at training step $t$, in terms of the difference in their unshaped returns:

$$\forall \pi : \mathbb{E}_{\pi_t}[G'_t] - \mathbb{E}_\pi[G'_t]$$
$$= \mathbb{E}_{\pi_t}[G] - \phi(h_t) + \gamma^N\mathbb{E}_{\pi_t}[\phi(h_{t+N})] - \mathbb{E}_\pi[G] + \phi(h_t) - \gamma^N\mathbb{E}_\pi[\phi(h_{t+N})] \quad (25)$$
$$= \mathbb{E}_{\pi_t}[G] - \mathbb{E}_\pi[G] + \gamma^N(\mathbb{E}_{\pi_t}[\phi(h_{t+N})] - \mathbb{E}_\pi[\phi(h_{t+N})]).$$

If one side of Equation 25 is nonnegative, the other side must also be nonnegative, thus:

$$\mathbb{E}_{\pi_t}[G'_t] \geq \mathbb{E}_\pi[G'_t] \iff \mathbb{E}_{\pi_t}[G] \geq \mathbb{E}_\pi[G] - \gamma^N(\mathbb{E}_{\pi_t}[\phi(h_{t+N})] - \mathbb{E}_\pi[\phi(h_{t+N})]) \quad (26)$$

Because $\pi_t$ was output by the RL algorithm, $\phi(h_{t+N})$ is part of its sequence of $\phi$ that has already converged. So Lemma A.2 gives us $\mathbb{E}_{\pi_t}[\phi(h_{t+N})] < \phi(h_t) + \epsilon$. Meanwhile, because $\phi$ is a monotone increasing function of time, we have $\mathbb{E}_\pi[\phi(h_{t+N})] \geq \phi(h_t)$ for all $\pi$. Applying these inequalities to the lower bound in 26, we can bound how much worse $\pi_t$ can be than any other policy $\pi$ if $\pi_t$ achieves higher shaped return:

$$\mathbb{E}_{\pi_t}[G'_t] \geq \mathbb{E}_\pi[G'_t] \iff \mathbb{E}_{\pi_t}[G] \geq \mathbb{E}_\pi[G] - \gamma^N(\mathbb{E}_{\pi_t}[\phi(h_{t+N})] - \mathbb{E}_\pi[\phi(h_{t+N})])$$
$$> \mathbb{E}_\pi[G] - \gamma^N(\phi(h_t) + \epsilon - \phi(h_t)) \quad (27)$$
$$\geq \mathbb{E}_\pi[G] - \epsilon,$$

where at the final line we used the fact that $\gamma \leq 1$. Thus, if the RL algorithm arrives at $\pi_t^{*'}$ that maximizes return shaped by a bounded monotone increasing BAMPF at time $t > H$, then $\pi_t^{*'}$ must also approximately maximize return in the underlying MDP, i.e.,

$$\forall \pi : \mathbb{E}_{\pi_t^{*'}}[G'_t] \geq \mathbb{E}_\pi[G'_t] \implies \forall \pi : \mathbb{E}_{\pi_t^{*'}}[G] > \mathbb{E}_\pi[G] - \epsilon. \quad (28)$$

We finally plug in the actual optimal policy for the original MDP $\pi^*$ in the right hand side to get our approximate-optimality-preserving result:

$$\mathbb{E}_{\pi_t^{*'}}[G] > \mathbb{E}_{\pi^*}[G] - \epsilon. \quad (29)$$

$\square$

*Remark.* This proof relies on all episodes being the same length. To handle MDPs with variable episode lengths, when an episode ends early all remaining BAMPF rewards that would have been added from the rest of the episode should be added at the terminating step. See Appendix H for an example.

## B  EXAMPLES OF PSEUDO-REWARD VALUE SIGNALLING

### B.1  PURE VOO SIGNAL

These pseudo-rewards help when $\bar{V}_O^*$ has a large influence on $\bar{Q}^*$ but the RL algorithm's (implicit or explicit) value estimate is misaligned with it. This often happens when there is significant prior information of the relative values of reaching states in $M$, which the algorithm is not initialized with, thus the pseudo-reward is often quite task or domain-specific. E.g., RL agents may underestimate value near the goal before first discovering the goal reward, or overestimate it near the icy cliff edge before first experiencing a fall. Attractive or repulsive $\bar{V}_O^*$ signal can correct RL algorithms that under or overestimate VOO, respectively.

#### B.1.1  ATTRACTIVE VOO SIGNAL

These terms help where the RL agent *underestimates* the value of getting to certain states, by rewarding it for reaching them. A common example is *goal proximity*-based reward shaping (Ng et al., 1999; Ghosh et al., 2018; Lee et al., 2021; Ma et al., 2022); which rewards each step of progress towards a goal in problems where the true reward is only at the goal itself. The goal location varies across MDPs in $p(M)$ but is fully observable from the initial state, therefore going towards it yields no information and thus does not increase $\bar{V}_I^*$, but is Bayes-optimal because it maximizes $\bar{V}_O^*$. An RL algorithm that is not initialized with this prior information would not prioritize going towards the goal over any other state, and would have to discover the sparse reward there through blind trial and error. Shaping rewards compensate for this, incentivizing behavior that approaches the goal.

Another common example with the same underlying mechanism is rewarding points scored in points-based-victory games like Pong. But if winning is not purely points-based, this is not necessarily good signal for $\bar{V}_O^*$; e.g., Clark & Amodei (2016) found an agent learned to crash itself to maximize points, when the true goal was to place first in the race.

#### B.1.2  REPULSIVE VOO SIGNAL

Psuedo-rewards based on repulsive $\bar{V}_O^*$ signal help when the agent would take suboptimal actions because it overestimates the Value of Opportunity, again due to missing domain knowledge, by penalizing behavior going to states with lower $\bar{V}_O^*$. Examples are *negative surprise* or *information-cost*-based IM (Berseth et al., 2019; Eysenbach et al., 2021) which give negative rewards based on the unpredictability of the states and transitions experienced. This is beneficial, assuming:

1. Unpredictable situations are undesirable in that domain, e.g. for driverless cars where it is dangerous to drive near other erratic vehicles, or robotic surgery, where it is dangerous to use unreliable surgical techniques with highly variable outcomes.

2. The RL algorithm does not apriori expect danger in unpredictable states and thus overestimates the $\bar{V}_O$ of exploring them; it could learn to avoid them by getting into accidents and receiving negative task rewards, but that would be very costly in the real world.

These rewards help RL algorithms by decreasing their estimates of the value of going to these dangerous states, better aligning them with their true value $\bar{Q}^*$, so they return to safety *before* getting into accidents.

More formally, negative surprise works under the assumption that *on the distribution of trajectories the agent actually experiences*, surprise almost always correlates well with negative outcomes. An example where this assumption wouldn't hold is if Times Square were a popular and safe destination for the driverless taxi, but the unpredictability of all the adverts were included in the surprise penalty. This measure of surprise correlates poorly with negative outcomes in trajectories through Times Square, so it would increase regret by making the agent unnecessarily reroute around it. In this problem, the signal for $\bar{V}_O^*$ must be more specific–only penalizing surprise with respect to things that could cause accidents, such as the positions of other vehicles.

## B.2 ATTRACTIVE VOI SIGNAL

Many intrinsic motivation terms are intended to reward behavior that gains valuable experience, reaching states with more valuable information in $h_t$. These are helpful for RL algorithms which don't account for the utility of gathering information (e.g., see section C), in problems containing information of significant value that is not immediately rewarding to collect. Note that different types of information are valuable in different types of problems, i.e., for different $p(M)$, and this determines which form of IM provides the most helpful signal for $\bar{V}_I^*$.

- *Prediction Error*-based IM (Schmidhuber, 1991; Pathak et al., 2017) rewards experiences that are predicted poorly by models trained on $h_t$. This helps when unpredictability given $h_t$ is good signal for the value of the information gained from the observation- thus, for IM based on dynamics models there must be *minimal stochasticity* (stochastic transitions are always unpredictable but yield no information) and in general *most information must be task-relevant* (so the information gained has value).

- *Count-based* IM (Bellemare et al., 2016; Burda et al., 2018; Lobel et al., 2023) provide a reward bonus for visiting states that have been visited less frequently, or in the case of large continuous state spaces pseudo-counts are used to group similar states into the same bucket. For example, Burda et al. (2018) observed dynamics prediction error failing in the 'noisy TV' problem, where a TV that changes channels randomly maximizes the intrinsic reward despite providing no information. This motivated their design of *RND*, which only predicts features of the current state as an estimate for how many times that state, or similar states, had already been visited. However, these types of IM can be counterproductive when the novelty of a state is a poor signal for how valuable it is to explore it, e.g. an "infinite TV problem" where the TV has infinite unique channels that provide useless information.

- *Entropy bonus* IM is proportional to the entropy of the MDP policy's action distribution (Szepesvári, 2010; Mnih, 2016; Haarnoja et al., 2017). This increases the estimated return of more stochastic $\pi$, so it can be understood as adding in the value of exploring a wider range of actions. This helps when RL algorithms get stuck in local maxima, but breaks down if the scale of the intrinsic rewards is too high, because overly random behavior is unlikely to reach interesting states, or could even be dangerous, e.g., if the increase in entropy when the robot overextends its joints outweighed the negative extrinsic rewards from the damage that does, it could encourage the robot to destroy itself. Therefore it must be carefully balanced with the scale and frequency of the extrinsic rewards (Hafner et al., 2023).

- *Mutual Information-based skill discovery* (Sharma et al., 2019; Warde-Farley et al., 2018; Eysenbach et al., 2018) rewards the agent based on the mutual information between the skills (temporally correlated sequences of actions) it learns and the resulting states. The higher this mutual information, the more diverse and controllable the skill set. It's a good signal for $\bar{V}_I^*$ in RL problems where this is a good measure for how useful the set of skills is for maximizing return, which also depends on the choice of representation used for the skills and states.

- *Information Gain* is a measure of the amount of information gained about the environment (Lindley, 1956). Info gain-based IM has led to successful exploration in RL (Sekar et al., 2020; Houthooft et al., 2016; Shyam et al., 2019); it is a good signal for $\bar{V}_I^*$ in problems where all information about the MDP is useful for maximizing return. But it could be a distraction in environments with many irrelevant dynamics to learn about, since the quantity of information gained would not always align with the value of that information.

## B.3 COMPOSITE SIGNAL

Finally, we analyze more complex pseudo-rewards that signal a combination of both $\bar{V}_I^*$ and $\bar{V}_O^*$.

### B.3.1 ATTRACTIVE VOI, REPULSIVE VOO

The *Empowerment* of an agent is typically measured as the mutual information $I(s'; a|s)$ between its actions and their effect on the environment (Klyubin et al., 2005; Salge et al., 2014a; Gregor et al., 2016). In prior work, this was generally understood as motivating agents to move to the states

of "maximum influence" (Salge et al., 2014b; Mohamed & Jimenez Rezende, 2015), e.g., the center of the room, or the junction of intersecting hallways. However, this does not always explain the full story. Mohamed & Jimenez Rezende (2015) found that in problems with predators chasing or lava flowing toward the agent, Empowerment motivates it to barricade itself from the lava or avoid the predators- even when this requires holing up in a tiny corner of the room. We can understand this by decomposing Empowerment into the sum of attractive $\bar{V}_I$ and repulsive $\bar{V}_O$ signals:

$$I(s'; a|s) = H(a|s) - H(a|s, s'), \tag{30}$$

where we can view $H(a|s)$ as adding attractive signal for $\bar{V}_I^*$, similar to Entropy Regularization, encouraging the exploration of different actions such as the barricade-placing action. Meanwhile $-H(a|s, s')$ adds repulsive signal for $\bar{V}_O^*$, similar to Negative Surprise, signalling that states where the agent is caught or engulfed (which results in the agent's actions ceasing to have any predictable effect on the environment) have low value, and thus should be avoided. Empowerment intrinsic motivation has mostly been tested in small finite environments, but this decomposition suggests its potential for lifelong learning in open-ended worlds, where it can encourage the exploration of a wide range of possibilities while staying out of danger.

Similar to Empowerment is Information Capture (Rhinehart et al., 2021), where the intrinsic reward is based on the expected entropy of the belief distribution (encouraging exploration) minus the entropy of the latent state visitation distribution (which is low when the agent has "stabilized" the environment by controlling it and creating a niche for itself).

### B.3.2 ATTRACTIVE VOI AND ATTRACTIVE VOO

Another example of a composite value signal is pseudo-rewards based on the principle of *optimism in the face of uncertainty*, e.g. Sorg et al. (2012); Qian et al. (2019). Rather than just rewarding novel experiences, they aim to reward actions where there is a high upper confidence bound on the possible value they could lead to. Thus, these types of pseudo-rewards encourage gaining information about the value of states and actions (signalling VOI), and/or reaching more valuable states (signalling VOO).

Reward bonuses for completing necessary subtasks for the first time, e.g. the first time successfully chopping wood as used in Crafter (Hafner, 2021), is one more example of a composite value signal. It adds both the value of discovering how to complete the subtask (because more wood will be needed) and the value of being one step closer to the ultimate objective of mining the diamond (one less woodblock needed to build a pick-axe). This adds the prior knowledge that in all worlds under $p(M)$, wood is a prerequisite for diamonds, and that it is valuable to learn how to gather new types of resources.

## C CERTAINTY-EQUIVALENT RL ALGORITHMS UNDERESTIMATE THE VALUE OF INFORMATION

Section 3.2 showed how we can understand many IM terms as encouraging and directing exploration through encoding the value of the resulting information. This type of IM is so popular because many commonly used RL algorithms under-estimate the value of information; we now introduce a model of these algorithms to formalize this within the BAMDP framework. Many RL algorithms, from policy-based methods like policy gradient (Sutton et al., 1999; Schulman et al., 2017) to value-based methods like Q-Learning (Watkins, 1989; Mnih et al., 2015), avoid the intractability of belief-space planning by acting as if beliefs are fixed. We formalize this objective with the Certainty-Equivalent[4] RL algorithm $\bar{\pi}^c$.

**Definition C.1** (Certainty-Equivalent RL Algorithm). At each step, the Certainty-Equivalent Algorithm $\bar{\pi}^c$ follows the MDP policy maximizing its estimated expected return under current beliefs:[5]

$$\bar{\pi}^c(\langle s_t, h_t \rangle) \in \arg\max_\pi \mathbb{E}_{b^c(h_t)}[V^\pi(s_t)], \tag{31}$$

---

[4]The idea of *certainty equivalent* solutions goes back to Simon (1956) and was also described by Duff (2002) as the *best reactive policy* with respect to $\bar{\pi}^c$'s beliefs

[5]Note that this is focused on exploitation- other than IM, ad-hoc methods such as epsilon-greedy or Boltzmann exploration may also be used to encourage exploration (Kaelbling et al., 1996).

where $b^c(\cdot)$ denotes how $\bar{\pi}^c$ interprets its experience, which could be anything from a distribution over world models maintained by updating a prior, to a point estimate of $Q^*$ maintained by training a randomly initialized neural net on batches sampled from $h_t$ (Mnih et al., 2015). For ease of notation we use $\pi_t$ and $a_t$ interchangeably, since $\pi_t$ is only used at step $t$ to output $a_t$.

For example, policy gradient algorithms like Reinforce sample actions from an MDP policy $\pi_\theta$, i.e., $\bar{\pi}^{\text{Reinforce}}(\langle s_t, h_t \rangle) = \pi_\theta(s_t)$, where $\pi_\theta$ is learnt by gradient updates towards maximizing the expected returns $\mathcal{R}(\tau)$ of the trajectories $\tau$ that it generates, i.e. $J(\theta) = E_{\tau \sim \pi_\theta}[\mathcal{R}(\tau)]$. The algorithm estimates $J(\theta)$ from environment interactions so far, i.e. $h_t$, so $\hat{J}(\theta) = \mathbb{E}_{b^c(h_t)}[E_{\tau \sim \pi_\theta}[\mathcal{R}(\tau)]]$ where $b^c(h_t)$ is concentrated on a point estimate of $J(\theta)$. If, as a model, we assume that policy gradient were to maximize $J(\theta)$ between each interaction, then we find that it matches the behavior of $\bar{\pi}^c$:

$$\arg\max_\theta \hat{J}(\theta) = \arg\max_\theta \mathbb{E}_{b^c(h_t)}[E_{\tau \sim \pi_\theta}[\mathcal{R}(\tau)]] = \arg\max_\pi \mathbb{E}_{b^c(h_t)}[V^\pi(s)]. \tag{32}$$

The Certainty-Equivalent algorithm effectively estimates the Q value as:

$$\hat{\bar{Q}}(\bar{s}_t, a) = \max_\pi \mathbb{E}_{b^c(h_t)}[R(s_t, a) + \gamma V^\pi(s_{t+1})]. \tag{33}$$

Comparing this to the optimal value $\bar{Q}^*(\bar{s}_t, a) = \mathbb{E}_{p(M|h_t)}[R(s_t, a) + \gamma \bar{V}^*(\langle s_{t+1}, h_{t+1} \rangle)]$, we see that, even assuming accurate beliefs $b^c(h_t) = p(M|h_t)$, the Certainty-Equivalent algorithm still misses value from the ability to learn from the information in $h$, because with $V^\pi(s_{t+1})$ it assumes it would still follow the same $\pi$ from step $t+1$, when in reality it would have updated it with the latest observation. E.g., if $s_b$ in Fig. 2 was observed to be empty, $\bar{\pi}^c$ would update $\pi$ to return to $s_w$ forever, resulting in higher return than if it kept following the same $\pi$ back to $s_b$ (see section I.2 for the calculations). Thus, adding pseudo-rewards signalling VOI can compensate for this underestimation, causing the algorithm to predict higher value for behavior leading to valuable information.

## D  CURIOSITY CASE STUDY

### D.1  BACKGROUND

The Curiosity intrinsic reward introduced by Pathak et al. (2017) is equal to the following error:

$$F(h_t) = \frac{\eta}{2}||\hat{e}(s_{t+1}; h_t) - e(s_{t+1}; h_t)||_2^2, \tag{34}$$

where $\hat{e}(s_{t+1}; h_t)$ is the output of a forward dynamics model, and $e(s_{t+1}; h_t)$ is a state encoder.

The encoder extracts features of the state that can be influenced by the agent, by minimizing the error of a learnt inverse dynamics model $i$ that predicts the action from the encoded state transition:

$$\hat{a}_t = i(e(s_t; h_t), e(s_{t+1}; h_t); h_t). \tag{35}$$

The forward dynamics model $d$ predicts the encoded next state given the current action and encoded state:

$$\hat{e}(s_{t+1}; h_t) = d(e(s_t; h_t), a_t; h_t), \tag{36}$$

and is continually trained to minimize the error of its prediction (as calculated in Equation 34). Curiosity uses this error to estimate the novelty of (and hence the added VOI from observing) states and transitions, assuming the error will decrease for the same observations as $d$ is updated. The Noisy TV problem occurs when the environment contains a screen that flips to random images every time a particular action is taken: watching TV doesn't increase VOI, but $d$'s error is highest and never decreases while watching. Thus, an RL algorithm trained with Curiosity is incentivized to produce a reward-hacking policy that maximizes the total shaped return by watching TV.

Burda et al. (2018) mitigate this issue with Random Network Distillation (RND) by predicting features of the current state rather than the next state. This would not get distracted by noisy transitions such as the TV flipping randomly between a few preset images, but it could get trapped forever in front of a TV that incremented through an infinite set of novel images. It could also neglect parts of the environment with *novel dynamics*, if the transitions are between less novel-looking states.

Curiosity's encoder and dynamics models are all trained on the history, so it could be converted to a BAMDP potential-based shaping function by directly using the error as the potential function

$\bar{\phi}(h_t) = ||\hat{e}(s_{t+1}; h_t) - e(s_{t+1}; h_t)||_2^2$. Though this would be fine for meta-RL, it wouldn't preserve optimality for RL because it is not monotonic. E.g., in the Noisy TV problem the prediction error always increases when watching TV and decreases upon looking away, because the prediction error of the totally random next image on the TV is higher than the error from exploring the rest of the environment.

## D.2 POTENTIAL-BASED SHAPING FUNCTION + BOUNDED MONOTONE BAMPF CONVERSION

We could try to capture the same type of dynamics prediction-based IM of Curiosity while preserving optimality, by splitting it into a potential-based shaping function plus a bounded monotone BAMPF. The PBSF can capture the part that is non-monotone: the novelty of the current state, while the BAMPF encourages observing novel dynamics that improve the dynamics model, by using its overall accuracy as the potential.

*Rewarding Novel States:* We could use a PBSF with $\phi(s)$ equal to the error of a *fixed model* $\hat{f}$'s prediction of fixed features $f$ of $s$ (similar to RND), encouraging visiting unfamiliar states:

$$\phi(s_t) = ||\hat{f}(s_t) - f(s_t)||_2^2. \tag{37}$$

Even if this error were higher at the TV, Ng et al. (1999)'s result guarantees that a policy can't maximize its total shaped return by staying there.

*Rewarding Novel Dynamics:* we could signal the total VOI at $h_t$ (see Definition 3.1) with the reduction in prediction error of the forward dynamics model compared to its initial error at $h_0$, which we call *Accuracy* for brevity. Setting $\phi(h_t)$ to this would encourage the agent to seek novel transitions to improve its predictions. We calculate the errors over a *fixed set of transitions* $\mathcal{T}$, so that Accuracy is bounded and should generally be expected to monotonically increase:

$$Acc(h_t) := \frac{1}{|\mathcal{T}|} \sum_{(s,a,s') \in \mathcal{T}} ||\hat{e}(s'; h_0) - e(s'; h_0)||_2^2 - ||\hat{e}(s'; h_t) - e(s'; h_t)||_2^2 \tag{38}$$

To guarantee that it monotonically increases, we can set $\bar{\phi}(h_0) = Acc(h_0)$ and make subsequent $\bar{\phi}(h_t)$ the exponentially smoothed maximum accuracy so far:

$$\bar{\phi}(h_{t+1}) = 0.5\bar{\phi}(h_t) + 0.5 \max\left(\bar{\phi}(h_t), Acc(h_{t+1})\right). \tag{39}$$

Even if TV-watching transitions were in set $\mathcal{T}$, $\bar{\phi}(h_h)$ could not increase indefinitely by any significant amount while watching TV, so the agent would not be incentivized to watch TV forever. The total intrinsic reward would simply be the sum of the PBSF and BAMPF:

$$\gamma(\phi(s_{t+1}) + \bar{\phi}(h_{t+1})) - (\phi(s_t) + \bar{\phi}(h_t)). \tag{40}$$

To prevent stagnation, every few batches we could update $\hat{f}$ (e.g., by training it to minimize its error over states visited in the new batches), and we could refresh $\mathcal{T}$ (e.g., with transitions that the dynamics model predicted the most poorly in the new batches). What matters for avoiding reward hacking is that the RL agent's choice of policy $\pi$ for the next episode can't affect $\hat{f}$ *within the episode* (so $\phi(s)$ remains a function of only the current MDP state), and that $\mathcal{T}$ is fixed for enough episodes for $\bar{\phi}(h)$ to converge.

## D.3 NUMERICAL DEMONSTRATION

We concretely demonstrate the conversion of Curiosity (Pathak et al., 2017) to an optimality-preserving PBSF + bounded monotone increasing BAMPF with a simple numerical experiment. We recreate the Noisy TV problem with the MDP shown in Fig. 7, with three available actions: *left*, *right*, and *watch TV*. The left and right actions deterministically transport the agent into the adjacent state in that direction (looping back to $S0$ and $S7$ when no such state exists). The *watch TV* action has no effect in all states except $S1$ and $S2$, where it randomly transports the agent to either $S1$ or

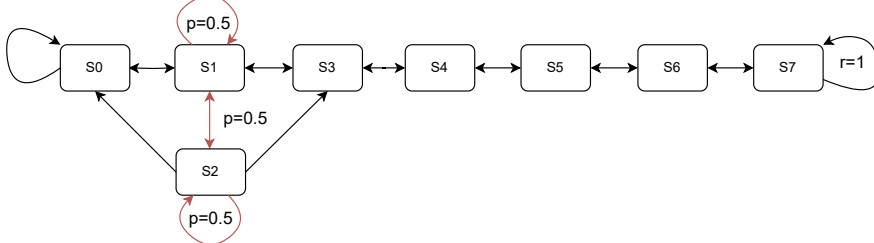

Figure 7: A discrete MDP recreating the Noisy TV Problem. Transitions are deterministic and rewards are 0 except where labeled explicitly. Taking the *watch TV* action from state $S1$ or $S2$ results in a stochastic transition to either state (red arrows).

$S2$ with equal probability. Thus, randomly flipping between $S1$ and $S2$ corresponds to watching a Noisy TV. The initial state is $S0$ and the only reward is obtained at $S7$.

No state feature encoding is needed for this small discrete state space, so we learn a forward dynamics model $d(s_t, a_t; h_t)$ directly on the states based on all the observed transition counts:

$$\hat{s}_{t+1;h_t} = d(s_t, a_t; h_t) := \arg\max_{s'} N_{h_t}(s, a, s'), \tag{41}$$

where $N_{h_t}(s, a, s')$ denotes the number of $(s, a, s')$ transitions observed so far. After observing any transition only once this model will predict it correctly- except for watching TV at states $S1$ or $S2$, where it will have a 50% chance of being correct no matter how many more observations it makes.

The states are represented as integers between 0 and 7, so we calculate the prediction error by taking the absolute distance. Thus, our Curiosity intrinsic reward is:

$$F(h_t) = \frac{\eta}{2}|\hat{s}_{t+1;h_t} - s_{t+1}|. \tag{42}$$

Training a tabular $\epsilon$-greedy Q-learning agent (episode length 8, $\gamma = 0.95$), we find that with $\eta \geq 4$, the agent reliably converges on a policy that watches TV rather than collecting the rewards at $S7$. Fig. 8a (purple curve) shows it achieves almost no real return, while Fig. 8b shows it achieves high shaped return, which must be almost entirely from the Curiosity rewards. Fig. 8d shows it estimates lower value for going right at state $S3$ (exploring further from $S0$) than left (back to the TV).

To convert Curiosity into a form that preserves optimality, we split it into a PBSF with $\phi(s)$ equal to the error of a fixed model's prediction of features of s (similar to RND (Burda et al., 2018)) to encourage visiting more novel states, plus a BAMPF with bounded monotone $\bar{\phi}(h)$ equal to the maximum accuracy of the dynamics model on a fixed set of transitions, to encourage observing more novel dynamics.

For the PBSF component, we again use the identity function as the feature encoding of the state $f(s_t) = s_t$, and for our fixed prediction model we use the following:

$$\hat{f}(s_t) = \begin{cases} f(s_t), & \text{if } s_t = 0 \\ f(s_t) - 1, & \text{if } 0 < s_t < 4 \\ f(s_t) - 3, & \text{otherwise} \end{cases} \tag{43}$$

and again use the absolute distance for the error, so our PBSF potential is $\phi(s_t) = |\hat{f}(s_t) - f(s_t)|$. Here $\hat{f}$ represents a model that is already very "familiar" with the initial state $S0$, somewhat familiar (but still making an error) with the closest states $S1, S2, S3$, but equally unfamiliar (i.e., making the same large error) with all other states.

For the bounded monotone BAMPF component, we used fixed set $\mathcal{T}$ of transitions from states $S1$-$S5$, including multiple TV-watching transitions. This represents a possible set of less well-predicted transitions collected in the intermediate stages of exploration. Again skipping the encoding and using absolute distance, we calculate Accuracy with $d$ (the same dynamics model used for

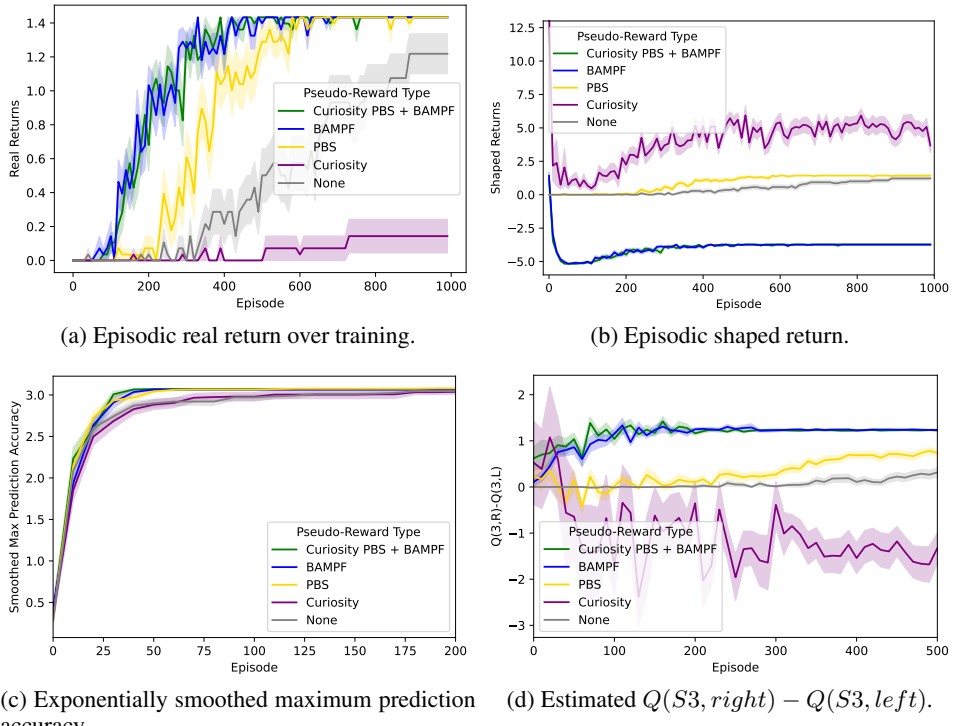

(a) Episodic real return over training.

(b) Episodic shaped return.

(c) Exponentially smoothed maximum prediction accuracy.

(d) Estimated $Q(S3, right) - Q(S3, left)$.

Figure 8: Curiosity intrinsic rewards (purple) cause the agent to learn a reward-hacking policy that maximizes its shaped return by watching TV, at the expense of the true return. Meanwhile, converting Curiosity into the form of a potential-based shaping function plus a bounded monotone BAMPF (green) improves exploration without resulting in a suboptimal policy. Each component individually (yellow and blue) also helps learning and preserves optimality. The mean and standard error over 20 training runs is plotted for each condition.

Curiosity), as:

$$Acc(h_t) := \frac{1}{|\mathcal{T}|} \sum_{(s,a,s')\in\mathcal{T}} |d(s,a;h_0) - s'| - |d(s,a;h_t) - s'|. \tag{44}$$

We finally set $\bar{\phi}(h)$ to the exponentially smoothed maximum accuracy so far following Equation 39; Fig. 8c shows how it evolves over training. We scale the BAMPF by 5 to get it into the same order of magnitude as the extrinsic reward function.

Fig. 8a shows how using this converted form of Curiosity (green curve) helped the Q learning agent discover how to reach the rewards at $S7$ more quickly, without incentivizing a suboptimal policy. Each component used individually (blue and yellow curves) also helps learning and preserves optimality, as expected. Fig. 8d shows how they increase the Q learning agent's estimate of the value of exploring further (going right) from state $S3$ long before any real rewards have been received, whereas without any pseudo-rewards (grey curve) it doesn't estimate that exploring further has any more value than going back until it actually starts reliably achieving the real rewards at $S7$.

Thus, we have converted the problematic curiosity IM term to a pseudo-reward with the same exploratory benefits, but without the costs of reward-hacking.

## E   MORE RELATED WORK

**Value of Information** The classical notion of the value of information originates in decision theory (Howard, 1966). Early work in metareasoning for tree search considers the utility of the information resulting from a computation (Russell & Wefald, 1989; 1991). Dearden et al. (1998) first applied the concept to RL, computing an approximation to the value of information to help select the Bayes-optimal action. They upper bound the "myopic value of information" for exploring action $a$ by the

expected Value of Perfect Information, i.e. the expected gain in return due to learning the true value of the underlying MDP's $Q^*(s, a)$ given prior beliefs. Chalkiadakis & Boutilier (2003) described BAMDP Q values as including the value of information—defined as the value of the change in beliefs quantified by its impact on future decisions—but they do not derive an expression for it. Ryzhov & Powell (2011) define the value of information of pulling a bandit arm as the expected resulting increase in the believed mean reward of the best arm, and derive an exact expression for bandits with exponentially distributed rewards.

**Reward Shaping and Meta-RL** Meta-Learning has been used to learn intrinsic motivation and reward shaping functions for RL (Zheng et al., 2018; Zou et al., 2019; Alet et al., 2020; Li et al., 2021; Zou et al., 2021), but so far few existing works use reward shaping or intrinsic motivation *to guide the meta-learner itself* (Zhang et al., 2021) and the preservation of optimal RL algorithms has not previously been investigated.

**Learning Awareness** The idea of planning through learning has appeared across RL such as in Bayesian RL and meta-RL (Thrun & Pratt, 1998; Duan et al., 2016; Finn et al., 2017; Mikulik et al., 2020; Jackson et al., 2024) and multi-agent RL (Foerster et al., 2017; Cooijmans et al., 2023).

## F    LEVER BAMPF DETAILS

We used the DQN implementation from Allen et al. (2021), with the default hyperparameters except `n_steps_init=20` and `decay_period=100` due to the shorter time horizon used. This agent uses epsilon-greedy exploration, with $\epsilon = 1$ for the first 20 steps, decaying linearly to 0.05 over the next 100 steps (according to the schedule $1 - 0.95 * \min(0.01(t - 20), 1.0)$). The *None* condition with no pseudo-rewards relied purely on this epsilon-greedy behavior for exploration. In this lifelong setting, the network is updated at every step with a batch of transitions from the full trajectory so far. The value of $\gamma$ used by both DQN and to define the BAMPF was 0.9. The Entropy Bonus intrinsic reward was calculated as $-10H(A)$ where $H(A)$ is the entropy of the categorical distribution of the last 10 lever pulls.

## G    BERNOULLI BANDITS META-RL DETAILS

We run the Bernoulli Bandits environment and A2C implementation in Gymnax (Lange, 2022a), keeping all the default hyperparameter values except for changing the number of pulls in an episode to 10. The discount factor used for the BAMPF is 0.8, matching the discount factor used by A2C.

Following Wang et al. (2016), we define the regret of an RL algorithm as $\sum_{t=1}^{T} \mu^* - \mu_{a_t} = \sum_{t=1}^{10} 0.9 - \mu_{a_t}$, where $\mu^*$ is the expected reward of the optimal arm and $\mu_{a_t}$ is the expected reward of the arm chosen at time $t$, so always pulling the optimal arm yields regret 0, and only pulling it half of the time (performance at random chance) has regret 4.

No rescaling was performed on either the BAMPF or the non-BAMPF pseudo-reward functions, since they were already of the same order of magnitude as the extrinsic rewards.

To handle the finite time horizon, we followed the approach of Grześ (2017) and set $\phi(h_{10}) = 0$ at the final (10th) step of each RL algorithm's trajectory, which ensures that the $\phi$ still sum to a constant over the trajectory to preserve optimality, and intuitively represents the fact that the final BAMDP state has no value, since the agent cannot collect any more rewards from it.

## H    MOUNTAIN CAR DETAILS

We run Mountain Car implemented in Gymnax (Lange, 2022a) using the PPO implementation from (Lange, 2022b), keeping all the default hyperparameter settings. The MountainCar environment itself has discount factor 1 so the reported return is simply the sum of the rewards in each episode, but the PPO algorithm uses discount factor $\gamma = 0.99$ so this was the discount used for both the potential-based shaping and the BAMPF. To handle the finite episode lengths, we again follow the approach of Grześ (2017) for the potential-based shaping, setting $\phi(s_t) = 0$ when $s_t$ is the last step in an episode, ensuring that it preserves optimality within each episode. For the BAMPF, at

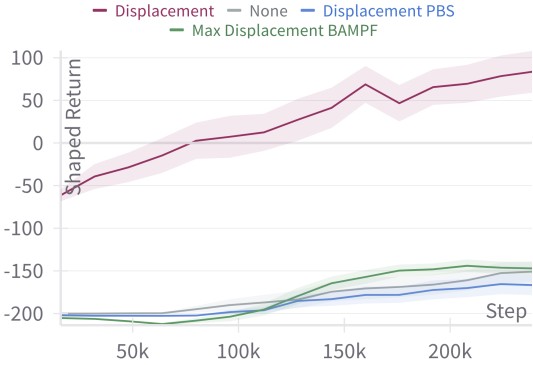

Figure 9: The (undiscounted) shaped episodic return for PPO trained with each type of pseudo-reward (mean and standard error of 10 seeds).

the last step in each episode we multiply $\phi(h_t)$ by $\gamma^{200-t_e}$ (where $t_e$ is the within-episode time-step and the maximum episode length is 200). This corresponds to adding up all the remaining BAMPF rewards that would have been added between that time-step and the final step if the episode hadn't been truncated (where in the remaining steps the history remains frozen at $h_t$ since no new experiences are collected). Our formula for exponential smoothing ($\alpha = 0.5$) on the maximum displacement is:

$$M_t = 0.5\max(M_{t-1}, |-0.5 - s_{xt}|) + 0.5M_{t-1}, \tag{45}$$

where $M_t$ denotes the exponentially smoothed maximum displacement at training step $t$, $s_{xt}$ is the x position of the car at time $t$ and $-0.5$ is the x position of the lowest point between the two hills. Note that this is still bounded and monotone increasing after smoothing. Our PPO implementation runs 16 copies of the environment in parallel; we maintain a separate $M_t$ for each copy (whenever an episode ends, $M_t$ is carried over to the next episode in that environment copy) and plotted the mean over the 16 values of $M_t$ in Fig. 5c.

For each form of pseudo-reward, we scale it by a constant to be on the same order of magnitude as the environment rewards. For the BAMPF and PBS, we scaled it by 10, and for Displacement (non-PBS version) we scaled it by 4. Note that whether Displacement preserves optimality is sensitive to this scaling factor, but the BAMPF version preserves optimality no matter the scale.

## I CATERPILLAR PROBLEM ANALYSIS

### I.1 BAYES-OPTIMAL POLICY VALUES

In section 2.3 we describe the behavior for the Bayes-optimal algorithm: for large enough $\gamma$, $\bar{\pi}^*$ should check $s_b$ first, then stay there forever if it find food, or else return to $s_w$ forever. Here we work through the math with $\gamma = 0.95$

From $\bar{s}_0$, the optimal RL algorithm $\bar{\pi}^*$ would take the action maximizing expected return under its knowledge $h_0$. If $\bar{\pi}^*$ were to stay and $eat$ at $s_w$, then $h_1$ would contain no more information than $h_0$, so $\bar{\pi}^*$ would make the same choice again and $eat$ forever. In that case, it would get a return of $\frac{21}{1-\gamma} = 420$.

If $\bar{\pi}^*$ were instead to first $go$ to $s_b$ and try to $eat$ there, its expected return would be:

$$-5 + \gamma\mathbb{E}[\bar{Q}^*(\langle s_b, h_1 \rangle, eat)], \tag{46}$$

where the first term is the energy cost of traveling. Now to calculate the second term we take an expectation over $p(M)$, i.e., take the weighted sum of the returns of an optimal RL algorithm in the presence and absence of food at $s_b$:

$$\mathbb{E}[\bar{Q}^*(\langle s_b, h_1 \rangle, eat)] = 0.1\frac{150}{1-\gamma} + 0.9(0 - 5\gamma + \gamma^2\frac{21}{1-\gamma}) = 637, \tag{47}$$

where the first term is the return if it finds food and thus eats at $s_b$ forever, and the second is the return if it finds no food and then goes back to eat at $s_w$ forever. Plugging this into 46, we get a total

expected return of 600. Since this is greater than 420, the Bayes-optimal RL algorithm $\bar{\pi}^*$ would follow this strategy.

## I.2 CERTAINTY-EQUIVALENT ALGORITHM VALUE CALCULATIONS

In section C we describe how the Certainty-Equivalent RL algorithm would act in the caterpillar MDP example. Here we go through the full calculations.

Algorithm $\bar{\pi}^c$, assuming it had the correct prior $p(M)$, would estimate the values of following various $\pi$ as follows:

- $\pi_b$ goes to the bush and stays there: $E_{p(M)}[V^{\pi_b}(s_w)] = -5 + 0.1 \times 150 \frac{\gamma}{1-\gamma} = 280$

- $\pi_{alt}$ alternates between the plants: $E_{p(M)}[V^{\pi_{alt}}(s_w)] = -5 \frac{1}{1-\gamma} = -100$

- $\pi_w$ eats at the weed forever: $E_{p(M)}[V^{\pi_w}(s_w)] = 21 \frac{1}{1-\gamma} = 420$; and it would go from $s_b$ so $E_{p(M)}[V^{\pi_w}(s_b)] = -5 + \gamma 420 = 394$

- $\pi_{eat}$ always eats wherever it is, so $E_{p(M)}[V^{\pi_{eat}}(s_w)] = 21 \frac{1}{1-\gamma} = 420$ and $E_{p(M)}[V^{\pi_{eat}}(s_b)] = 0.1 \times 150 \frac{1}{1-\gamma} = 300$

Because $\pi_w$ gets the highest estimated value, $\bar{\pi}^c$ would choose to follow it, thus never learning about the bush and staying at the weed forever.

As an example of $\bar{\pi}^c$ underestimating its own value, take its estimate of its value of eating from $\langle s_b, h_0 \rangle$, i.e. if it were at the bush with no additional information. It assumes it would follow the best MDP policy under current information at the next step no matter what it found at $s_b$, which is still $\pi_w$, giving estimate:

$$\hat{\bar{Q}}^c(\langle s_b, h_0 \rangle, eat) = E_{p(M)}[R(s_w, eat) + \gamma V^{\pi_w}(s_b)] = 0.1 \times 150 + 394\gamma = 369 \qquad (48)$$

However, this significantly underestimates the value. If $\bar{\pi}^c$ ate at $s_b$ and found no food, it would update to $\pi_w$ to go and eat at $s_w$, and if it did find food it would update to a $\pi$ that continues eating at $s_b$. This corresponds to a much higher true value:

$$\bar{Q}^c(\langle s_b, h_0 \rangle, eat) = 0.1 \frac{150}{1-\gamma} + 0.9(-5\gamma + \frac{21\gamma^2}{1-\gamma})) = 637 \qquad (49)$$

