# OpenReview forum: "BAMDP Shaping: a Unified Framework for Intrinsic Motivation and Reward Shaping"
_ICLR.cc/2025/Conference — ICLR 2025 Poster_

### Official Review · Reviewer_SDzM · 2024-10-17

**Soundness:** 2
**Presentation:** 2
**Contribution:** 3
**Rating:** 6
**Confidence:** 3

**Summary:**

This paper investigates the design of pseudo-rewards (e.g., intrinsic motivation, reward shaping) and when it can lead to unwanted side effects. To achieve this, the authors first cast the problem into a meta-learning problem by considering the effect of pseudo-rewards at the scale of RL algorithms using the BAMDP framework.
By decomposing value functions into value of information/opportunity (VOI/ VOO), the authors developed a new way to classify different
The authors also showed that the optimality of an RL algorithm is preserved if and only if the pseudo-rewards belong to the class of potential-based reward shaping in the corresponding BAMDP.

**Strengths:**

The idea of studying pseudo-rewards in the meta-learning setting is very original and interesting. The main theoretical result which says that pseudo-rewards can preserve optimality of an RL algorithm if and only if they are potential-based is very insightful and might be useful for designing new generations of exploration methods.

**Weaknesses:**

My main concern is that most of the performance guarantee results are proven at the scale of meta-learning, and it is not clear to me whether this actually solves the “reward-hacking” problem at the scale of individual MDPs, which is the main motivation of the paper. For example, if only 1 out of 100 of the MDPs suffers from the noisy TV problem, then the optimal RL algorithm in this case might simply “sacrifice” that MDP by using a novelty-based approach to explore more efficiently in the rest of the MDPs if the rewards from these MDPs far outweigh the ones from the noisy TV MDP.
Another minor weakness is that the paper mostly focuses on the invariance properties of the class of BAMPFs, and it remains unclear what are the advantages/disadvantages of choosing a specific BAMPF, which limits the practical relevance of the paper.

Also, as this paper is more theory focused, I feel like some parts of the paper are not clearly/rigorously written (see questions).

**Questions:**

1. In section 3.2, the authors draw connections between IM, VOI, and VOO. However, it’s not clear to me how IM is related to VOI or VOO quantitatively. Also, what are the definition of attractive/repulsive signals?

2. On line 323, “In environments containing finite information…”, what is the definition of information for an environment?

3. I’m not sure if Def 4.2 is well-defined. Consider the same single-state MDP/potential function defined on line 364, except that now there are infinitely many levers. In this case, whether the potential function is settling is going to depend on the policy, that is, the potential function is settling if the policy has non-zero probabilities on finitely many levers (independent of time), and will not be settling otherwise.

4. In figure 3, what is None? Does it make use of any exploration method?

---

> ### Author Response · Authors · 2024-11-20
> **Response to SDzM**
>
> We thank reviewer SDzM for their thoughtful review and helpful feedback. We are glad that they found our approach to studying pseudo-rewards to be very original and interesting, and our theoretical results insightful.
>
> >most of the performance guarantee results are proven at the scale of meta-learning, and it is not clear to me whether this actually solves the “reward-hacking” problem at the scale of individual MDPs
>
> Thank you for flagging that this aspect of our contribution is unclear. We have updated our language in section 4.3 in the revised manuscript to make this clearer- our result there is proven for the regular RL setting. We also added an experiment in this section in the MountainCar environment to empirically demonstrate how reward-hacking at the MDP level is avoided with a BAMPF. We show a BAMPF pseudo-reward helping PPO learn an optimal goal-reaching policy more quickly, whereas a similar non-potential based IM term initially helped exploration but eventually led to a reward hacking policy that avoids the goal to collect more pseudo-rewards.
> See the revised manuscript and the overall response for more details of the new experiments.
>
> > it remains unclear what are the advantages/disadvantages of choosing a specific BAMPF, which limits the practical relevance of the paper.
>
> We appreciate this feedback for making our paper more practical. We have included discussion about how we chose effective BAMPFs for the new experiments (see the overall response or sections 4.2.2 and 4.3.1 in the revised manuscript) which we hope will be a helpful example for practitioners seeking guidance on how to design BAMPFs for their own settings.
>
> **Questions:**
> >how IM is related to VOI or VOO quantitatively
>
> We discuss further in Appendix C how RL algorithms can underestimate the value of information in their base objective, which can explain how many IM terms help exploration by adding approximations of VOI back into the objective. But we agree that this section is mostly qualitative; it is generally not possible to calculate the true VOI and VOO, but it can still be a useful framework for understanding and designing IM terms. The Mountain Car (section 4.3.1) and Bernoulli Bandits (section 4.2.2) experiments we added provide more concrete examples of how the framework can guide the design of and understand the effects of IM.
>
> >what are the definition of attractive/repulsive signals?
>
> We briefly explained attractive and repulsive signals after the definition of VOO, but we see how that could be easy to miss so we now reiterate it in the caption of Table 1. An attractive signal provides (usually positive) rewards to correct value underestimation, and a repulsive signal provides (usually negative) rewards to correct value overestimation
>
> *Questions 2 and 3:*
>
> Thank you for flagging the lack of clarity and precision in the wording here. By environments with finite information, we meant to intuitively describe environments in which potential functions measuring the amount of information gathered cannot increase indefinitely. We did intend “settling” to describe a phenomenon that depends on both the RL agent’s trajectory and the potential function, but have now reworked the theory in this subsection to be simpler and more direct. Realizing that every example of Settling we thought of involved a bounded monotone potential function, we cut out the intermediate definition of Settling and the policy-dependence. Instead we directly prove that BAMPFS based on bounded monotone potentials preserve approximate optimality for any RL agent (see section 4.3 of revised manuscript)
>
> >In figure 3, what is None? Does it make use of any exploration method?
>
> *None* corresponds to the DQN agent without any pseudo-rewards. The DQN agent we ran in this demonstration uses epsilon-greedy exploration, with epsilon=1 for the first 20 steps, then decaying to 0.05 over the next 100 steps. We have clarified this in the caption and text and added these details to the appendix.

---

> > ### Comment · Reviewer_SDzM · 2024-11-20
> >
> > I thank the authors for the detailed response.
> >
> > However, Theorem 4.3 does not really address my concern. My question is that whether we can guarantee the performance of the optimal BAMDP policy $\bar{\pi}^{\*}$ for an individual MDP (i.e., what can we say about $\mathbb{E}_{\bar{\pi}^{\*}}[G|\mathcal{M}]$?) In contrast, the theorem relates optimal policies with or without BAMPF shaping within a given MDP.
> >
> > Regarding attractive/repulsive signals, I was asking for mathematical definitions of them, as I found these concepts difficult to understand without precise definitions.
> >
> > Another minor point: to define a monotone function $\phi(h)$, you'll have to first define an order over $h$.

---

> ### Author Response · Authors · 2024-11-22
>
> We thank reviewer SDzM for considering our response.
>
> (We realized that latex sometimes doesn't display properly unless broken into separate lines, hence the multiple line breaks in our reply here:)
> > Theorem 4.3 does not really address my concern. My question is that whether we can guarantee the performance of the optimal BAMDP policy for an individual MDP
>
> Apologies for misunderstanding your concern. It is indeed possible that for some $M$ in the support of $p(M)$, $\mathbb{E}_{\bar{\pi}^{*}}[G|\mathcal{M}]$ is significantly lower than the expected performance over $p(M)$.
>
> To say more about $\mathbb{E}_{\bar{\pi}^{*}}[G|\mathcal{M}]$ would involve doing some type of worst-case robustness analysis.
>
> We do not address this problem in our paper, but this is not what we mean by reward hacking. We address reward hacking as defined in [1] as rewards that “mislead the agent into learning suboptimal policies”.
> This is where an RL algorithm is **incentivized** to generate a policy $\pi^{*'}$
>
> that maximizes expected shaped return $\mathbb{E}_{\pi^{*’}}[G|\mathcal{M}’]$,
>
> but achieves suboptimal expected real return $\mathbb{E}_{\pi^{*’}}[G|\mathcal{M}]$.
>
> Our theorems state what forms of pseudo-reward can be added to a BAMDP (4.2)/MDP(4.3) without changing which RL algorithms/MDP policies maximize expected BAMDP/MDP return, respectively, and thus can't cause reward-hacking by definition.
>
>
> >Regarding attractive/repulsive signals, I was asking for mathematical definitions of them
>
> Thank you for flagging that these concepts are still unclear. Here is a more precise mathematical definition: a pseudo-reward is an attractive signal towards a metric M if the reward increases in the metric, i.e., $f(h1) \geq f(h2) \iff M(h1) \geq M(h2)$ (where e.g., f is state coverage and M is the number of states visited) and it is a repulsive signal if it decreases in the metric, i.e., $f(h1) \leq f(h2) \iff M(h1) \geq M(h2)$ (e.g., f is the joint angle violation penalty and M is the angle of a robot’s joint). This isn't a full classification of how pseudo rewards could incorporate metrics, just an intuitive typology for many reward shaping or IM terms that are designed to encourage or discourage specific behaviors or outcomes. We have updated the manuscript to make this clearer.
>
> > to define a monotone function $\phi(h)$, you'll have to first define an order over $h$.
>
> We thank the reviewer for pointing this out. The sequence of histories is ordered by time, i.e., for all training steps $t_1<t_2$, either $\phi(h_{t_1}) \leq \phi(h_{t_2})$ for a monotone increasing $\phi(h)$, or $\phi(h_{t_1}) \geq \phi(h_{t_2})$ for a monotone decreasing $\phi(h)$. We have updated the manuscript to specify this explicitly.
>
> Thanks again for taking the time to engage with our work, and please let us know if you have any more questions
>
> [1] Ng, Andrew Y., Daishi Harada, and Stuart Russell. "Policy invariance under reward transformations: Theory and application to reward shaping." Icml. Vol. 99. 1999.

---

> > ### Comment · Reviewer_SDzM · 2024-11-23
> >
> > Thank you for the response and clarifications; however, I'm still uncertain about the performance guarantee.
> > (also sorry about having to break lines because the equations are not rendering properly)
> >
> > If I'm understanding correctly, while BAMPF preserves $\mathbb{E}_{\bar{\pi}^*,\mathcal{M}}[G]$ ,
> >
> > it might still change how $\mathbb{E}_{\bar{\pi}^*}[G|\mathcal{M}]$ behaves for each environment.
> >
> > Now, if we cannot guarantee the individual performance $\mathbb{E}_{\bar{\pi}^*}[G|\mathcal{M}]$, is it not the case that there might exist a BAMPF that makes the performance arbitrary bad for some environments (i.e., misleading the agent into learning suboptimal policies)?
> >
> > Note that this does not contradict with Theorem 4.3, as the theorem only states that the performance is preserved for the *optimal* policies (with/without shaping) of a given environment. The problem here is that the Bayes-optimal RL algorithm might not find the optimal policies for some environments.

---

> ### Author Response · Authors · 2024-11-23
>
> We thank the reviewer for taking the time to consider our responses again, and appreciate their patience in explaining their concern-- it is a challenge to write clearly about concepts at both the RL and meta-RL level within the same work, and this discussion is valuable for highlighting ways that we could improve the clarity of our writing.
>
> It's possible that some confusion may be arising from the fact that we're discussing reward hacking in meta-RL, but the Noisy TV problem[1,2] (and most well-known cases of reward hacking) is an instance of reward hacking in regular RL. This is where the pseudo-rewards cause RL algorithms (which may or may not be Bayes-optimal BAMDP policies themselves) to converge on MDP policies that maximize shaped return $\mathbb{E}_{\pi}[G|\mathcal{M^\prime}]$,
>
> but perform suboptimally with respect to the true return $\mathbb{E}_{\pi}[G|\mathcal{M}]$ in an individual MDP. E.g., in the Noisy TV problem, the eventual policy $\pi$ learnt by the RL algorithm $\bar{\pi}$ maximizes the shaped return by watching TV, while this behavior doesn't maximize the true return. **This empirically documented problem, which matches the definition of reward hacking from [4], is the major motivation of the paper, which we address with theorem 4.3.**
>
> Now we return to the discussion about reward hacking in meta-RL (which is the purview of theorem 4.2).
>
> > is it not the case that there might exist a BAMPF that makes the performance arbitrary bad for some environments?
>
> It cannot make performance arbitrarily bad - if we guide a meta-RL agent with BAMPF pseudo-rewards and the resulting Bayes-Optimal RL algorithm $\bar{\pi}^*$ performs worse in one MDP, it must perform correspondingly better in other MDPs such that the lost performance balances out, preserving the optimal expected performance $\mathbb{E}_{\bar{\pi}^*,\mathcal{M}\sim p(\mathcal{M})}[G]$. Thus, the learnt RL algorithm still *behaves rationally with respect to the true distribution of MDPs* $p(\mathcal{M})$, and this is not an instance of reward hacking where the pseudo-rewards mislead the meta RL agent to optimize for a different reward distribution.
>
> >The problem here is that the Bayes-optimal RL algorithm might not find the optimal policies for some environments.
>
> It is true that a Bayes-optimal BAMDP policy might not find an optimal MDP policy for some MDPs. But this is not a problem caused by BAMPFs or any sort of reward hacking - this is a consequence of how the optimality of an RL algorithm (and the meta-RL objective) is defined in terms of its expected performance over $p(\mathcal{M})$, rather than worst-case performance. Reimagining the mainstream RL objective to optimize for best or worst-case performance has been explored, e.g., [3], and it is an interesting question but orthogonal to our work.
>
>
> We hope that these explanations help resolve the reviewer's uncertainty, but please let us know if there are remaining concerns.
>
> [1] Burda, Yuri, et al. "Exploration by random network distillation." arXiv preprint arXiv:1810.12894 (2018).
>
> [2] Mavor-Parker, Augustine, et al. "How to stay curious while avoiding noisy tvs using aleatoric uncertainty estimation." International Conference on Machine Learning. PMLR, 2022.
>
> [3] Veviurko, Grigorii, Wendelin Böhmer, and Mathijs de Weerdt. "To the Max: Reinventing Reward in Reinforcement Learning." arXiv preprint arXiv:2402.01361 (2024).
>
> [4] Ng, Andrew Y., Daishi Harada, and Stuart Russell. "Policy invariance under reward transformations: Theory and application to reward shaping." Icml. Vol. 99. 1999.

---

> > ### Comment · Reviewer_SDzM · 2024-11-25
> >
> > As it currently stands, I think the only thing can be said about BAMPF is that it will preserve the performance of RL algorithms, which is definitely interesting in its own way, but I don't see how this can allow us to "design new pseudo-reward terms in this form to avoid unintended side effects".
> > If I'm understanding correctly, the present work only states that we can shift the unintended side effects from one environment to another while preserving the expected performance when averaged over environments.
> >
> > In addition, I believe there is an issue with the proof of Theorem 4.3. In Equation 23, $G$ should be replaced by $G_t$, meaning that the performance guarantee only holds after $t$ steps, not for earlier steps (and therefore not for $G$ as a whole). To fully characterize the performance difference, it is necessary to account for $G_{<t}$, where $\phi(h_t)$ may not have converged yet.
> > Furthermore, the $H$ in lemma A.2 may be policy-dependent (i.e., $H(\pi)$), and whether this can be upper bounded (i.e., $\exists H^* > H(\pi) \forall \pi$) might depend on the class of policies considered.

---

> ### Author Response · Authors · 2024-11-26
> **Response 1/2**
>
> We thank the reviewer for taking the time to elaborate on their concerns again.
>
> **Addressing questions and concerns about Theorem 4.2 (meta-RL setting):**
>
> > BAMPF ... will preserve the [expected] performance of RL algorithms... but I don't see how this can allow us to "design new pseudo-reward terms in this form to avoid unintended side effects"
>
> When we talk about unintended side effects we mean decreasing expected performance, which is the standard metric used in meta-RL. (Note that if we're in a setting where optimal RL algorithms always find the optimal MDP policy, then after adding a BAMPF the resulting RL algorithm will still always find the optimal MDP policy)
>
> >If I'm understanding correctly, the present work only states that we can shift the unintended side effects from one environment to another while preserving the expected performance when averaged over environments.
>
> Performing worse in some MDPs and better in others is not a side-effect of BAMPFs: even with no reward shaping, meta-RL generally learns RL algorithms (that may still be Bayes optimal) with this behavior. And while Theorem 4.2 doesn't guarantee performance in all individual MDPs, our work does show we can preserve optimal performance of MDP policies in all MDPs with Theorem 4.3.
>
> **Addressing questions and concerns about Theorem 4.3 (RL setting):**
>
> >In Equation 23, $G$ should be replaced by $G_t$
>
> We believe the reviewer has misunderstood our notation here. Apologies for not providing sufficient explanation, here is a full explanation which we will also add to the paper:
>
> $\mathbb{E}_{\pi}[G_t^{\prime}]$ refers to the expected discounted sum of both BAMPF rewards and real rewards that MDP policy $\pi$ achieves when rolled out at training step $t$, which varies with $t$ because the BAMPF varies over time.
>
> $\mathbb{E}_{\pi}[G]$ is the expected discounted sum of just the real rewards that $\pi$ achieves-- because the real reward function doesn't change over time, this return is constant with respect to any $\pi$.
>
> >the performance guarantee only holds after $t$ steps, not for earlier steps (and therefore not for $G$ as a whole)
>
> Our performance guarantee in Theorem 4.3 is for the performance of the MDP policy $\pi_t^{* \prime}$ learnt by the RL algorithm, $\mathbb{E}_{\pi^{*\prime}_t}[G]$ (i.e., the *test-time performance* of the learnt policy),
>
> which is different from all the rewards obtained by RL algorithm *during learning*, i.e., $\mathbb{E}_{\bar{\pi}}[G]$. So, Theorem 4.3 doesn't guarantee that RL algorithms will always achieve high returns throughout training, it **guarantees the optimality of the learnt MDP policy at test-time**. This is sufficient to address the problem of reward hacking both as defined by Ng et al. and observed empirically- when the RL algorithm converged on an optimal MDP policy for the shaped return (e.g., the TV watching policy) which is not optimal for the real return.
>
> Note that, though Theorem 4.3 doesn't address performance throughout training, Section 3 (value decomposition) provides principles and Section 4.3 provides empirical examples of how to design bounded monotone BAMPFs that do improve learning.
>
> > The $H$ in lemma A.2 may be policy-dependent, and whether this can be upper bounded might depend on the class of policies considered
>
> The sequence $\phi(h_t)$ depends on the *RL algorithm*, not any one MDP policy $\pi$, since the RL algorithm determines the history of experiences throughout learning $h_t$ (often through outputting a sequence of policies $\pi_1,\pi_2,...$). Thus, $H$ may certainly be algorithm dependent ($H(\bar{\pi})$). But as long as $H$ exists for any given RL algorithm (which we proved), we guarantee that it preserves optimality. Thus, we can guide any RL algorithm with a bounded monotone BAMPF while guaranteeing that optimality is preserved.

---

> ### Author Response · Authors · 2024-11-26
> **Response 2/2**
>
> **Looking over the reviewer's questions and concerns, it seems that it might be helpful to clarify the distinction between the meta-RL versus RL settings of our results. So we restate our theorems below, trying to make the differences as explicit as possible:**
>
> **Theorem 4.2: (meta-RL):** we can preserve $E_{p(M)}[ \mathbb{E}_{\bar{\pi}^{\ast}}[G|M] ]$: the optimal expected (over both the MDP distribution $p(M)$, and any stochasticity within each MDP's $T$ and $R$ functions) return of the RL algorithm $\bar{\pi}$ (i.e., *performance throughout learning*) learnt by any meta-RL agent, if we guide the meta-RL agent with BAMPF rewards.
>
> **Limitations:** doesn't guarantee worst-case performance of learnt RL algorithm in individual MDPs
>
> **Theorem 4.3: (regular RL)** we can preserve $\mathbb{E}_{\pi^*}[G]$: the optimal expected (just over any stochasticity in $T,R$) return in *all MDPS* of the MDP policy $\pi$ (i.e., *test-time performance of a fixed policy*) learnt by any RL algorithm, if we guide the RL algorithm with bounded monotone BAMPF rewards.
>
> **Limitations:** doesn't guarantee the rewards obtained throughout learning (but we provide principles and examples for designing BAMPFs to improve learning performance in sections 3 and 4.3)
>
> We will further clarify the scope and limitations of each theorem in the paper, and thank the reviewer for bringing this area of improvement to our attention. But to reiterate, Theorem 4.3 does fully address the problem of reward hacking, both as defined by Ng. et al, and as it is widely observed empirically (e.g., the Noisy TV problem). We believe that this in itself is a contribution worthy of publishing at ICLR. (Theorem 4.2 addresses reward-hacking for meta-RL, which is not yet well-known since reward-shaping for meta-RL is still a nascent technique, but was recently discussed in e.g., [1].)
>
> We hope that this has helped clear up any misunderstandings of our work, but please do let us know if there are remaining concerns, and again we appreciate the reviewer's continued patience and thoughtful engagement.
>
> [1] Zhang, Jin, et al. "Metacure: Meta reinforcement learning with empowerment-driven exploration." International Conference on Machine Learning. PMLR, 2021

---

> > ### Comment · Reviewer_SDzM · 2024-11-27
> >
> > Thank you once again for the very detailed explanations regarding the scope and limitations. Upon further reflection, I believe my concerns largely stem from the narrative presented in the abstract and the introduction. Currently, the authors motivate the work through examples of how bonus rewards can lead to unintended side effects in the regular RL setting, which may lead the readers into thinking that the proposed solution can address these problems in the regular RL setting rather than the meta RL setting.
> > I would like to emphasize that this in no way diminishes the authors' contribution, which, as I noted in my initial review, is both original and interesting. However, I do believe that the writing could be improved to more accurately reflect the contributions.
> >
> > As my major concerns have been addressed, I will increase my score.
> >
> > Regarding Lemma A.2: since the proof of Theorem 4.3 requires that there exists an $H$ such that the inequality holds for any $\pi_t^{*\prime}$, you need to show that $H(\bar{\pi})$ ($\forall\bar{\pi}$ optimal at $t$) can be bounded by $H$. Currently, the lemma only proves that $H(\bar{\pi})$ exists, which is not enough because $\phi(h_t)$ can converge arbitrarily slow for some $\bar{\pi}$.

---

> ### Author Response · Authors · 2024-11-28
>
> We thank the reviewer for another insightful response. We are glad to hear we have addressed their main concerns, while we greatly value their continued engagement in helping us improve our manuscript.
>
> > Currently, the authors motivate the work through examples of how bonus rewards can lead to unintended side effects in the regular RL setting, which may lead the readers into thinking that the proposed solution can address these problems in the regular RL setting rather than the meta RL setting.
>
> We do propose a solution to reward hacking in the regular RL setting (e.g., the Noisy TV problem)- the bounded monotone BAMPF in section 4.3 addresses this setting. But we appreciate the reviewer letting us know that the narrative could be improved. We see how it may be confusing since we start with our meta-RL contribution in section 4.2 before the regular RL setting in 4.3, but we did not thoroughly motivate or introduce the problem of reward shaping for meta-RL in the introduction or abstract. We will rework the introduction and abstract to make the narrative clearer and more representative, and we have already edited the discussion in the revised manuscript to make the contribution and limitations more explicit.
>
> > Regarding Lemma A.2:...you need to show that $H(\bar{\pi})$ ($\forall\bar{\pi}$ optimal at $t$) can be bounded by $H$.
>
> Thank you for further explaining your concern. To clarify, do you mean "$\forall \pi$ optimal at $t$"? In Theorem 4.3 we only discuss optimal MDP policies $\pi$ (where $\pi_t^{*\prime}$ is an optimal MDP policy for the shaped rewards at time $t)$, not optimal RL algorithms $\bar{\pi}$ (since this theorem is for the regular RL setting). The convergence speed is fully specified by the RL algorithm $\bar{\pi}$, and we don't have a different horizon for each optimal MDP policy at time $t$. This is because every RL algorithm $\bar{\pi}$ generates its own sequence of training histories $h_1,h_2,...h_t$ and thus its own bounded monotone sequence $S^{\bar{\pi}}_t=\phi(h_1),\phi(h_2),...\phi(h_t)$. Though the transitions and rewards in $h$ are typically collected by rolling out MDP policies $\pi_1,\pi_2,...$, note that each $\pi$ is generated by the RL algorithm based on the history ($\pi_t=\bar{\pi}(h_t)$) and only used for a finite time e.g., a batch of 100 episodes, before being updated.
>
> But we think we understand the reviewer's concern now. To prove Theorem 4.3, we use Lemma A.2 to bound the difference in $\phi(h_{t+N})$ between two MDP policies $\pi_1,\pi_2$ if they were rolled out from the same point in training $h_t$:
>
> $E_{\pi_1}[\phi(h_{t+N})] - E_{\pi_2}[\phi(h_{t+N})] < \epsilon$
>
> But if $\pi_1,\pi_2$ are arbitrary policies not necessarily generated by RL algorithm $\bar{\pi}$ during training, then potential $\phi$ at the new histories $h_{t+N}$ they induce may not be part of the same sequence $S_t^{\bar{\pi}}$. So even if  $S_t^{\bar{\pi}}$ has converged, $E_{\pi}[\phi(h_{t+N})]$ might still change by more than $\epsilon$ for these hypothetical MDP policies. Thus, Lemma A.2 can only be applied for MDP policies $\pi$ that were actually generated by the RL algorithm during training. Did we correctly understand the reviewer's concern?
>
> In light of this, Theorem 4.3 can instead guarantee *the ordering of all policies tried by the RL agent is preserved*. The main conclusions are unchanged: preserving the ordering still prevents reward-hacking since the RL algorithm cannot generate a policy $\pi_t$ that gets higher shaped return but lower real return than any other policy it tries. (This is much closer to our original result in this section, before we restructured our theory to bypass the definition of Settling following reviewers' suggestions).
>
> We are very grateful for the reviewer's care and attention to detail with our theory. We will revise our manuscript accordingly, taking extra care to check all the details. Please let us know if this addressed the concern!

---

> > ### Comment · Reviewer_SDzM · 2024-11-28
> >
> > > Thus, Lemma A.2 can only be applied for MDP policies $\pi$ that were actually generated by the RL algorithm during training. Did we correctly understand the reviewer's concern?
> >
> > Correct.
> >
> > I agree that the main conclusion remains unchanged, but I believe the statement of the theorem can be made more precise.

---

> ### Author Response · Authors · 2024-12-02
> **Thank you for the really thoughtful and helpful discussion**
>
> We thank the reviewer for the clarification on this. We agree, and will update our theorem statement to make it more precise as described in our previous response.
>
> Thanks so much again for your thoughtful responses and high level of engagement throughout this discussion period, which has been extremely helpful for improving our paper!

---

### Official Review · Reviewer_AbDy · 2024-10-21

**Soundness:** 3
**Presentation:** 2
**Contribution:** 3
**Rating:** 8
**Confidence:** 3

**Summary:**

The authors propose a unifying decision-making framework that incorporates the concepts of intrinsic motivation, reward shaping, and potential functions. In particular, they propose a Bayes-adaptive MDP (BAMDP) framework that can reformulate intrinsic motivation and reward shaping as BAMDP potential-based shaping functions, thereby avoiding some of the known pitfalls of these methods. They show that this MDP framework satisfies important theoretical properties that make it possible to find an optimal policy, and provide an empirical experiment in support of this.

**Strengths:**

This paper provides a theoretically-sound framework that is able to incorporate the essence and advantages of intrinsic motivation and/or reward shaping while avoiding some of the known pitfalls of these methods. The decomposition of the value function into two components is insightful and interesting. The paper is generally well-written, and the technical concepts are explained in a relatively clear manner.

**Weaknesses:**

My biggest concern with this paper is the lack of empirical experiments. While it is reasonable to expect that a theoretical paper such as this one will not have as many experiments, having a single experiment with only a single state is a bit underwhelming. In particular, it would be useful if the authors included at least one other, more complicated experiment that further illustrates the usefulness of their framework. Ideally, the additional experiment(s) would show and contrast how the BAMDP avoids the reward-hacking (or similar) behavior that can occur with intrinsic motivation and/or reward shaping.

Similarly, it would be useful if the authors could provide a case study showing how a well-known, problematic instance of intrinsic motivation or reward shaping can be converted into a BAMDP potential-based shaping function. I understand that the authors provided a generic, brief overview of this process in lines 315-320, however having a specific example of a well-known, documented instance of reward-hacking (or similar) that would be alleviated by their framework would greatly solidify the arguments presented in this paper.

Finally, the limitations of the framework could use more discussion. In particular, it would be useful to have an example of intrinsic motivation and/or reward shaping (if any) that cannot be converted into a BAMDP potential-based shaping function. This would better characterize the usefulness and applicability of the framework. Ultimately, any potential user of the framework needs to be able to infer from this paper under what conditions the framework should or should not be used.

Some minor comments:
1) Figure 1 is hard to read. In particular, the caption should be more descriptive, it is not clear which individual plots belong to ‘a)’ vs ‘b)’, and the color scale should have a title.
2) Figure 2 is overcrowded (making it hard to read) and needs to be simplified.
3) Each item in Table 1 should have a reference/citation attached to it, especially since some of the items are not mentioned in the main text.

**Questions:**

Is there a type of intrinsic motivation and/or reward shaping that cannot be converted into a BAMDP potential-based shaping function? In other words, do these items have to conform to some sort of form or adhere to a certain set of assumptions?

---

> ### Author Response · Authors · 2024-11-20
> **Response to AbDy**
>
> We thank reviewer AbDY for their thoughtful and constructive comments. We are glad that they found our paper to be well-written and our theoretical results to be insightful and interesting.
>
> **Experiments:**
>
> We agree that the paper would be strengthened with more experiments and less simplistic settings. We have added two more experiments to section 4; in both settings, as suggested, we contrast a BAMPF with other forms of pseudo-reward, showing how it speeds up learning while avoiding reward-hacking.
>
> - In the Mountain Car environment, which is episodic and involves a continuous two-dimensional state space. We demonstrated a BAMPF pseudo-reward signalling VOI helps PPO learn a goal-reaching policy more quickly while preserving optimality. In contrast, regular potential-based shaping did not provide any improvement, and adding similar non-potential based rewards initially helped exploration, but eventually led to a reward hacking policy that avoids the goal to collect more pseudo-rewards
>
> - In the Bernoulli bandit meta-learning problem introduced by [1] we show that a BAMPF helps the A2C meta-learner find the optimal RNN-based RL agent more quickly by discouraging over-exploration, while the non-BAMPF version of the same pseudo-reward led to a reward-hacking RL agent that over-exploits.
>
> We also explain how we design the potentials and understand the effects of our pseudo-rewards using the principles from our framework in section 3 (value signaling and decomposition). See the revised manuscript and the overall response for more details of these new experiments.
>
> **Case Study:** This is a great point. We have added a case study describing how one could use our framework to convert Curiosity[2], a well-known IM term that suffers from the Noisy TV problem, into a form that preserves optimality and thus alleviates the Noisy TV problem (section 4.3.2 in the revised manuscript)
>
> **Limitations:** Thank you for this suggestion. The BAMPF potential function includes all pseudo-rewards that are functions of the algorithm’s learning history, which captures virtually any form of reward shaping or intrinsic motivation. This guarantees the optimal RL algorithm will be preserved if we use this to shape meta-RL. However, to provide reward hacking guarantees for regular RL we also require the potential to be bounded and monotone, which is more limited. This includes rewards based on the entire history like state coverage or information gain, but excludes rewards purely based on the latest actions like prediction error (the predictability of the latest observations) or entropy bonus (the randomness of the latest policy). We show in the Curiosity case study that we can still convert some of these to preserve optimality, by expressing them as the sum of a PBS function and a bounded monotone BAMPF. But in general, the conversion process is manual and may not be immediately obvious. We have added discussion of these limitations to section 4.3.2 and the Discussion section.
>
> We thank the reviewer for their feedback for making the figures and tables more legible and informative. We fleshed out the captions, added more space between the subfigures and labeled the colorbar. We have simplified figure 2 and believe it is now much easier to read. We have added citations to every term in Table 1.
>
> [1] Wang, Jane X., et al. "Learning to reinforcement learn." arXiv preprint arXiv:1611.05763 (2016).
>
> [2] Pathak, Deepak, et al. "Curiosity-driven exploration by self-supervised prediction." International conference on machine learning. PMLR, 2017.

---

> ### Comment · Reviewer_AbDy · 2024-11-20
>
> I thank the authors for their response. The added empirical validation has increased my confidence in the proposed framework and I have updated my score accordingly. As per the case study, what was presented in the updated draft is too brief and not very helpful to the reader if they are not familiar with Curiosity. I encourage the authors to include an appendix that expands upon the brief paragraph included in section 4.3.2, such that it presents the conversion process explicitly, with equations and, ideally, some simple numerical experiment (note that I will not decrease my current score if the appendix is not included, however I am happy to further increase my score if it is).

---

> ### Author Response · Authors · 2024-11-27
> **Thank you for the valuable suggestion**
>
> We thank reviewer AbDy for their fast response. We really appreciate the suggestion to provide more background on Curiosity and include a simple experiment-- we agree this would make the case study more helpful and concrete. Thus, we have expanded on it in Appendix D in the revision we just uploaded: in D.1 we introduce Curiosity more thoroughly, in D.2 we go through the conversion explicitly with equations, and finally in D.3 we present a simple experiment where we recreate the Noisy TV problem, and show that the converted version of Curiosity is not susceptible to it. We hope this in-depth example is helpful for readers looking to convert their own problematic intrinsic motivation/reward shaping terms into a form that preserves optimality. Thank you for helping us further improve our manuscript!

---

> > ### Comment · Reviewer_AbDy · 2024-11-27
> >
> > I thank the authors for the inclusion of Appendix D in the latest version of the paper. This appendix has made the case study, and the overall paper, more concrete and easier to understand and as such, I have increased my score.

---

> > > ### Author Response · Authors · 2024-12-02
> > >
> > > Perfect; thanks again for an insightful review, and for the constructive and timely responses during the discussion period!

---

### Official Review · Reviewer_jfJX · 2024-11-04

**Soundness:** 3
**Presentation:** 2
**Contribution:** 2
**Rating:** 6
**Confidence:** 3

**Summary:**

This paper studies the problem of reward modeling based on intrinsic motivation (IM) and reward shaping. Although IM generalizes reward-shaping functions in the standard MDP setting, it can still be expressed as reward-shaping in the BAMDP setting. Leveraging the fact that potential-based reward shaping functions (PBSFs) preserve optimal policies under any shaping, an extension to BAMDPs is proposed. This leads the authors to introduce BAMPFs, which are potential functions at the level of BAMDP: there, the state becomes the MDP history or a learning algorithm. A result shows that asymptotically stationary potential functions preserve optimal policies. This hints at how one should model the reward function in complex tasks such that the learned policy does not exhibit undesirable side-effects.

**Strengths:**

The paper is well-motivated: previous literature is introduced properly, and the positioning of this work is clear.

The interpretation of IM as reward shaping in BAMDPs is novel, to the best of my knowledge.
Also, it serves a practical objective, namely, reward modeling for complex RL tasks, a critical problem that is particularly challenging.

**Weaknesses:**

The paper is sometimes hard to follow:
-  too long sentences
-  technical words are used already from the abstract without being introduced or explained. It took me several readings to understand what is going on in this paper.

I am unfamiliar with most of the related work (just the BAMDP part) so I cannot fully assess the quality of this submission. I know that BAMDPs are hard to work with, in terms of interpretability and tractability, but proposing experiments on a bandit problem seems insufficient for a mature publication.

**Questions:**

- The "RL algorithms" paragraph in Sec. 2.1 should appear together with BAMDPs from Sec. 2.3

- Did the authors consider extensions of their method to "realistic" environments? Do they already have an algorithm that leverages this BAMPF idea in deep RL?

---

> ### Author Response · Authors · 2024-11-20
> **Response to jfJX**
>
> We thank reviewer jfJX for their helpful comments. We are glad that they find our work to be novel and well-motivated; we agree that it addresses a critical challenge in reward modeling.
>
> We agree that experiments in more complex and realistic settings would be helpful for demonstrating our theory. We added two deep RL experiments to section 4:
>
> - In Mountain Car, which involves a two-dimensional continuous state space and simulates motion under Newton’s laws, we demonstrated a BAMPF pseudo-reward signalling VOI helps PPO learn a goal-reaching policy more quickly while preserving optimality. In contrast, regular potential-based shaping did not provide any improvement, and adding similar non-potential based rewards initially helped exploration, but eventually led to a reward hacking policy that avoids the goal to collect more pseudo-rewards
>
>  - In the Bernoulli Bandit meta-RL setting introduced by [1] we show that a BAMPF helps the A2C meta-learner find the optimal RNN-based RL agent more quickly by discouraging over-exploration, while the non-BAMPF version of the same pseudo-reward led to a reward-hacking RL agent that over-exploits.
>
> See the revised manuscript and the overall response for more details of these new experiments.
>
> We appreciate the feedback on the clarity of the writing. To make the paper easier to follow, we have gone through the paper and broken up long sentences, and reduced the use of technical jargon where possible. We also moved the “RL algorithms” paragraph to section 2.3 as suggested.
>
> [1] Wang, Jane X., et al. "Learning to reinforcement learn." arXiv preprint arXiv:1611.05763 (2016).

---

> > ### Author Response · Authors · 2024-11-27
> >
> > Thank you again for your constructive review; we hope we have addressed your concerns with our responses and revisions, particularly the new experiments in more complex settings. As the deadline to upload a revision approaches, please let us know if you have any further questions or concerns!

---

> > > ### Comment · Reviewer_jfJX · 2024-11-27
> > >
> > > Sorry for my late response and thank you for considering my concerns and addressing my suggestions.
> > >
> > > After having gone through all other reviews and discussions, I think the authors did a good job accounting for our comments to improve their submission. I increase my score accordingly.

---

> > > > ### Author Response · Authors · 2024-12-02
> > > >
> > > > We thank the reviewer for taking the time to consider our responses and revisions, we are glad to hear that they have addressed the reviewer's concerns!

---

### Official Review · Reviewer_kw7i · 2024-11-05

**Soundness:** 3
**Presentation:** 3
**Contribution:** 3
**Rating:** 8
**Confidence:** 3

**Summary:**

The paper presents a theoretical framework that allows us to study both intrinsic motivation and reward shaping together in a unified way. This affords us the analysis of well-known intrinsic motivation and reward shaping methods, and their pathologies. They show that both intrinsic motivation and standard reward shaping can be lifted onto the Bayes-adaptive MDP level, where they together form different ways to do reward shaping for the BAMDP. Then prove that any standard IM and reward shaping method that corresponds to a potential-based shaping in the BAMDP should lead to Bayes-optimality in the underlying problem.

**Strengths:**

- The BAMDP value decomposition in Lemma 3.2 is to my knowledge novel. It is deceptively simple, yet seems to afford important results.

- Table 1 presents a great overview of how different IM and reward shaping methods fit together under the decomposition.

- The proposed framework of BAMPFs improve our mathematical understanding of IM and reward shaping in RL, and their known pathologies.

- Theoretical results appear novel and of high quality.

- Paper is written clearly and it is easy to read.

**Weaknesses:**

- The empirical result is illustrative, more than demonstrative. A bigger-scale experiment would have been better. Specifically, I would appreciate a longer-horizon example. It can be tabular.

- The theoretical results are heavy on definitions, to the point of sounding circular. Specifically, the Theorem 4.3. is not really surprising, and follows from applying the definition of potential-based shaping to BAMDPs. Although I agree that it is more complicated than the standard MDP case.

- Plots in Figure 3 are very low resolution to the point of being blurry.

**Questions:**

Overall, I think this is a good paper, providing good insights into IM and reward shaping. However, I think it would have benefitted from showing its impact with studying more examples of different IM and reward shaping terms, demonstrating how we can study their pathologies.
Can you present more examples demonstrating the impact of your results?

---

> ### Author Response · Authors · 2024-11-20
> **Response to reviewer kw7i**
>
> We thank reviewer kw7i for their generally positive review of our paper. We are glad that they find our results to be novel, high quality and important.
>
> **Experiments:** We agree that more examples, particularly larger-scale and longer-horizon empirical examples, would be helpful to demonstrating the impact of our theoretical results.  We added two more experiments to section 4; in both we use our theory to understand the pathologies of various pseudo-rewards, and to design BAMPF pseudo-rewards that improve learning while avoiding reward hacking.
>
> - In Mountain Car, which involves a 2D continuous state space and episodes up to 200 steps long, we trained PPO for 250k steps. We demonstrated a BAMPF pseudo-reward which signals VOI helps PPO learn a goal-reaching policy more quickly. In contrast, regular potential-based shaping did not provide any improvement, and similar non-potential based rewards initially helped exploration, but eventually led to a reward hacking policy that avoids the goal to collect more pseudo-rewards
>
> - In the Bernoulli Bandit meta-RL setting introduced by [1] we used A2C to learn an RNN-based RL agent over 4000 updates/256k episodes. BAMPF helps the A2C meta-learner find the optimal RNN-based RL agent more quickly by discouraging over-exploration, while the non-BAMPF version of the same pseudo-reward led to a reward-hacking RL agent that over-exploits.
>
> We hope these additional experiments better demonstrate the utility of our framework. See the revised manuscript and the overall response for more details of these new experiments.
>
> **Theory:** We appreciate the feedback about the conceptual clarity and definition-heaviness of our theory, and have done some restructuring which we hope will make it easier to follow. We modified our theorem on preserving MDP optimality (Theorem 4.3 in the revised manuscript) to bypass the intermediate definition of Settling, which we believe makes it clearer and more direct. We also expanded the BAMPF abbreviation in the statement of Theorem 4.3 (now 4.2 in the revised manuscript), making it less jargon-heavy. Though this result might not seem surprising, it is not currently widely appreciated: to the best of our knowledge, we are the first to design potential-based reward shaping for meta-RL. Given it is natural and intuitive, we believe it is particularly valuable to share with the community.
>
> Thank you for pointing out the blurriness of the plots, we have remade them in higher resolution.
>
> [1] Wang, Jane X., et al. "Learning to reinforcement learn." arXiv preprint arXiv:1611.05763 (2016).

---

> > ### Comment · Reviewer_kw7i · 2024-11-26
> >
> > Thank you for the response. Overall, these address my main issues. I genuinely like this paper and I believe we need more papers in this line increasing our understanding and also unifying seemingly different approaches under a framework. It helps organise knowledge and thinking. I will increase my score.

---

> > > ### Author Response · Authors · 2024-11-27
> > >
> > > We are glad to hear we addressed your concerns, and greatly appreciate the encouraging feedback. Many thanks again for the thoughtful and constructive review!

---

### Author Response · Authors · 2024-11-20
**Overall Response**

We thank all the reviewers for their thoughtful responses. We are glad that they found our work to be novel (SDzM,kw7i, jfJX), insightful (AbDy, SDzM), theoretically sound (kw7i, AbDy) and addressing an important challenge (jfJX,kw7i). We also greatly appreciate the constructive feedback, which has helped us improve our paper. We have uploaded a revised manuscript incorporating the feedback, with changes highlighted in blue.

In particular, a common theme among reviewers was that larger-scale experiments in more complex domains would help demonstrate the practical impact of the theory. **We have added two experiments to section 4, showing BAMPFs improve learning efficiency while preserving optimality, whereas similar pseudo-rewards result in reward hacking**. These examples also demonstrate how our BAMDP value framework can inform the design of more effective pseudo-reward functions.

- We train PPO in Mountain Car, which has a 2D continuous state comprising the car’s position and velocity. The reward is -1 at each step; each episode lasts 200 steps or ends early if the car reaches the goal. The car starts in a valley, and to reach the goal on the hill it must build momentum by first moving up the opposite slope. Thus, displacement from the lowest point in the valley is an intuitive shaping reward to signal the VOO of being further uphill in either direction. We find this helps PPO reach greater displacements early in training (red curve in Fig. 5c), but eventually results in reward-hacking policies that collect more pseudo-rewards by avoiding the goal (Figures 5b and 7). Converting this to classic potential-based shaping by setting $\phi(s)$ to the displacement, we find it preserves optimality but doesn't help learning (blue curve in Fig.
5b), possibly because the further uphill the car gets, the more negative PBS rewards it soon receives when it rolls back. We can understand these failures as a result of the fact that displacement is too weak a signal for the true VOO in Mountain Car. Instead, we could signal VOI by rewarding the *maximum* displacement so far, which doesn't penalize temporary decreases in displacement. This value clearly depends on more than the current state, so isn't a valid MDP potential- but it *is* a valid BAMDP potential $\phi(h)$. It signals the VOI of the RL agent learning how to get further uphill; because the agent sometimes surpasses its maximum displacement by chance before learning how to consistently get so far, we apply exponential smoothing. Fig. 5c (green curve) shows the resulting BAMPF gets the agent to explore further more quickly, and Fig. 5b shows it learns to reach the goal and maximize true rewards sooner while avoiding reward-hacking.

- We train A2C in the Bernoulli Bandit meta-RL problem introduced by [1]. The MDPs in this domain have two arms, one with reward probability 0.1 and the other 0.9 (randomly assigned), and a budget of 10 pulls. At each step an RNN-based RL agent observes the previous arm pulled, the reward it produced, and how many pulls it has left, and chooses which arm to pull next. After 10 pulls, the episode ends and a new MDP is sampled. The A2C meta-learner continually updates the RNN agent to maximize expected return in its 10-step lifetime, i.e., to find the RL algorithm that optimally balances exploration and exploitation. Without any shaping, A2C gradually learns to generate RL agents that try fewer arms on average, i.e., avoiding over-exploration (Fig. 3b, grey curve). This shows that it initially over-estimates the VOI of trying more arms, so we add a BAMPF to correct its prior: setting $\phi(h)$ to the total pull count of the first arm that produced a reward, decreasing the relative perceived VOI of exploring the other arm. We find it helps A2C learn lower-regret RL agents more quickly, while still converging to an optimal agent (Fig. 3b, green curve). In contrast, when this pull count is added directly as a pseudo-reward, it causes the meta-learner to converge on an agent that over-exploits and never achieves the optimal regret (Fig. 3, purple curve).

See the revised manuscript for plots & further experimental details. We hope these additional experiments more concretely demonstrate the utility of our framework

In addition to the experiments, feedback from the reviewers on the writing has helped us improve the clarity of the paper in many ways. We detail how this has been done in individual responses to the reviewers.

**Updates:**
Thank you all for a very constructive discussion period! Following reviewer AbDy's valuable suggestion, we explicitly demonstrate the use of our BAMPF framework to convert a well-known problematic IM term to a form that avoids reward-hacking, making this part of our paper more concrete. Reviewer SDzM's thoughtful, detailed feedback helped us further clarify our narrative and make our theory more precise.

[1] Wang, Jane X., et al. "Learning to reinforcement learn." arXiv preprint arXiv:1611.05763 (2016).

---

### Meta-Review · Area_Chair_FPLH · 2024-12-20

**Metareview:**

BAMDP Shaping: A Unified Theoretical Framework for Intrinsic Motivation and Reward Shaping

Summary: This paper addresses the role of pseudo-rewards in reinforcement learning (RL) through the lens of Bayes-Adaptive Markov Decision Processes (BAMDPs). It presents a theoretical framework for understanding intrinsic motivation (IM) and reward shaping by analyzing how pseudo-rewards guide exploration through BAMDP state values, decomposed into Value of Information (VOI) and Value of Opportunity (VOO). The paper introduces BAMDP Potential-Based Shaping Functions (BAMPFs), which ensure pseudo-rewards preserve the optimality of RL algorithms and avoid reward hacking. The authors provide theoretical results showing that BAMPFs can preserve both RL and meta-RL optimality. Empirical demonstrations in the Mountain Car environment and Bernoulli Bandits domain highlight the practical utility of BAMPFs in improving learning efficiency and avoiding suboptimal exploration incentives while maintaining theoretical guarantees.

Comments: We received 4 expert reviews, with the scores 6, 6, 8, 8, and the average score is 7.0. The reviewers are generally positive about the technical contributions of this paper. In particular, the reviewers commented that the BAMDP decomposition into Value of Information (VOI) and Value of Opportunity (VOO) is novel and insightful, and provides a unified theoretical framework for intrinsic motivation and reward shaping. This approach also offers a practical method to design pseudo-rewards while avoiding common issues such as reward hacking. The reviewers appreciated the strong theoretical approach used by the authors to address this intuitive and challenging question. The paper is also well-written.

The reviewers have also provided some valuable suggestions to strengthen the paper. I am glad to see that the authors have already accommodated many of the suggestions into their revised version during the rebuttal period. However, there is one concern, about limited empirical results, which is the lack of larger-scale or longer-horizon examples. This concern is valid as it limits the ability to evaluate the clear practical impact of the framework. I recommend the authors to Include more experiments on MuJoCo style environments.

**Additional Comments On Reviewer Discussion:**

Please see the "Comments" in the meta-review.

---

### Decision · Program_Chairs · 2025-01-22

Accept (Poster)